**Investigation**

# High-recombining genomic regions affect demography inference based on ancestral recombination graphs

Jun Ishigohoka (ID),[1,*] Miriam Liedvogel (ID)[1,2,3,*]

[1]Max Planck Research Group Behavioural Genomics, Max Planck Institute for Evolutionary Biology, August-Thienemann-Straße 2, Plön 24306, Germany
[2]Institute of Avian Research, An der Vogelwarte 21, Wilhelmshaven 26386, Germany
[3]Department of Biology and Environmental Sciences, Carl von Ossietzky Universität Oldenburg, Ammerländer Heerstraße 114-118, Oldenburg 26129, Germany

*Corresponding author: Max Planck Research Group Behavioural Genomics, Max Planck Institute for Evolutionary Biology, August-Thienemann-Straße 2, Plön 24306, Germany. Email: ishigohoka@evolbio.mpg.de; *Corresponding author: Max Planck Research Group Behavioural Genomics, Max Planck Institute for Evolutionary Biology, August-Thienemann-Straße 2, Plön 24306, Germany. Email: liedvogel@evolbio.mpg.de

Multiple methods of demography inference are based on the ancestral recombination graph. This powerful approach uses observed mutations to model local genealogies changing along chromosomes by historical recombination events. However, inference of underlying genealogies is difficult in regions with high recombination rate relative to mutation rate due to the lack of mutations representing genealogies. Despite the prevalence of high-recombining genomic regions in some organisms, such as birds, its impact on demography inference based on ancestral recombination graphs has not been well studied. Here, we use population genomic simulations to investigate the impact of high-recombining regions on demography inference based on ancestral recombination graphs. We demonstrate that inference of effective population size and the time of population split events is systematically affected when high-recombining regions cover wide breadths of the chromosomes. Excluding high-recombining genomic regions can practically mitigate this impact, and population genomic inference of recombination maps is informative in defining such regions although the estimated values of local recombination rate can be biased. Finally, we confirm the relevance of our findings in empirical analysis by contrasting demography inferences applied for a bird species, the Eurasian blackcap (*Sylvia atricapilla*), using different parts of the genome with high and low recombination rates. Our results suggest that demography inference methods based on ancestral recombination graphs should be carried out with caution when applied in species whose genomes contain long stretches of high-recombining regions.

**Keywords:** demography inference; ancestral recombination graph; recombination rate; population genomics; nonmodel species; *Sylvia atricapilla*

## Introduction

Population history affects the patterns of genetic variation, and conversely observed genetic variation in genomes can be informative for demography inference. Some methods for demography inference are based on the ancestral recombination graph (ARG) (Li and Durbin 2011; Schiffels and Durbin 2014; Terhorst *et al.* 2017; Speidel *et al.* 2019), a structure that describes the full ancestries of sampled genomes (Griffiths and Marjoram 1997). It essentially consists of a series of marginal genealogical trees changing in shape and branch lengths along the chromosome, and their breakpoints effectively represent historical recombinations contributing to the sampled genomes (Fig. 1). The full ARG provides rich information on the population history (i.e. all coalescence and recombination events through time and mutations mapped on branches), making ARG-based methods a powerful population genomics approach to study evolutionary processes (Speidel *et al.* 2019; Stern *et al.* 2019; Hubisz *et al.* 2020; Schaefer *et al.* 2021; Wohns *et al.* 2022). In practice, however, ARG-based methods depend on inference of the ARG (Rasmussen *et al.* 2014; Mirzaei and Wu 2017; Kelleher *et al.* 2019; Speidel *et al.* 2019; Ignatieva *et al.* 2021; Wohns *et al.* 2022), or representations of underlying genealogies (Li and Durbin 2011; Schiffels and Durbin 2014; Terhorst *et al.* 2017), which in turn relies on observed mutations. Importantly, the presence of mutations representing an ARG branch depends on recombination and mutation rates. If an ancestral haplotype breaks by a recombination before accommodating mutations, the corresponding branch on the ARG is not represented by any mutations (Fig. 1b) (Hayman *et al.* 2023; Shipilina *et al.* 2023). Therefore, high recombination rates (relative to the mutation rate) makes it difficult to accurately infer the underlying ARG, limiting the performance of the ARG-based approach (Terhorst *et al.* 2017; Sellinger *et al.* 2020, 2021; Dutheil 2024).

Could recombination landscape in some organisms affect ARG-based demography inference? In humans, which is the most commonly used species for initial empirical application, such impact is presumably negligible (Li and Durbin 2011; Terhorst *et al.* 2017), because recombination rate in humans is low except for narrow recombination hotspots (Myers *et al.* 2010; Stevison *et al.* 2016). However, recombination landscapes with globally low rate are not universal to all organisms (Auton *et al.* 2013; Lam and Keeney 2015; Singhal *et al.* 2015; Baker *et al.* 2017). For example, recombination landscapes differ within vertebrates, which can be partially attributed to the presence and absence of PRDM9, a transcription

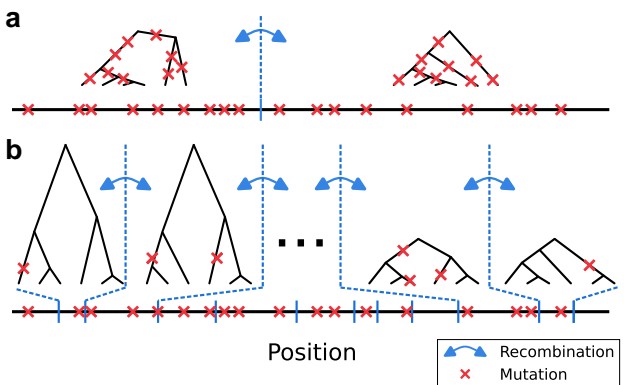

**Fig. 1.** The presence of mutations representing ARG branches depends on recombination rate. a) When recombination rate is moderately low, branches of ARG are represented by mutations. This allows inference of the underlying ARG based on observed mutations. b) When recombination rate is high, many branches of ARG are not represented by any mutations. We ask whether this affects ARG-based demography inference.

factor that localizes recombination hotspots at rapidly evolving binding DNA motifs (Oliver *et al.* 2009; Baudat *et al.* 2010; Myers *et al.* 2010; Baker *et al.* 2015; Paigen and Petkov 2018). Following multiple independent loss of PRDM9 across vertebrates (Cavassim *et al.* 2022), recombination hotspots shifted from motifs to genome features such as transcription start sites and CpG-islands (Auton *et al.* 2013; Singhal *et al.* 2015; Baker *et al.* 2017; Kawakami *et al.* 2017; Paigen and Petkov 2018). Recombination hotspots appear to be wider in organisms lacking PRDM9, such as birds (Singhal *et al.* 2015; Kawakami *et al.* 2017; Bascón-Cardozo *et al.* 2024), dogs (Auton *et al.* 2013), and percomorph fish (Baker *et al.* 2017), compared to PRDM9-dependent hotspots in primates (Durbin *et al.* 2010; Myers *et al.* 2010; Stevison *et al.* 2016). Differences in recombination landscape are also common among a wide range of organisms outside vertebrates (Choi *et al.* 2013; Lam and Keeney 2015; Smukowski Heil *et al.* 2015; Stukenbrock and Dutheil 2018; Palahí i Torres *et al.* 2023). In addition, the average recombination rate can be highly variable between chromosomes and species (Stapley *et al.* 2017; Martin *et al.* 2019; Bascón-Cardozo *et al.* 2024). This diversity in meiotic recombination could potentially impact modeling of local ARGs for demography inference in some organisms.

In this study, we ask how recombination landscapes affect ARG-based demography inference. To this end, we simulate genome data under simple demographic histories with various recombination maps and evaluate demography inference by different ARG-based methods. Specifically, we focus on two ARG-based methods, MSMC2 (Malaspinas *et al.* 2016; Wang *et al.* 2020) and Relate (Speidel *et al.* 2019), differing in the way the ARG is modeled. While Relate infers a series of marginal genealogies along the genome with their topology and branch lengths collectively representing the full ARG of the sample, MSMC2 models the distribution of the coalescence times between pairs of sampled sequences along the genome based on the sequentially Markovian coalescent (SMC, McVean and Cardin 2005). Using simulated data, we also examine whether masking high-recombining regions improves ARG-based demography inference in organisms with high-recombining regions, and whether recombination maps inferred by population genomics methods can be used to define high-recombining regions to mask for ARG-based demography inference. To demonstrate the relevance of our findings based on simulations, we translate our findings to empirical data of a species with wide high-recombining

genomic regions. Specifically, we use whole-genome resequencing (WGR) data and fine-scale recombination maps of a songbird species, the Eurasian blackcap (*Sylvia atricapilla*, "blackcap" hereafter), and contrast ARG-based demography inferences using genomic regions differing in recombination rates.

## Materials and methods
### Simulation
#### Recombination maps

To investigate the effect of high-recombining regions on demography inference, we prepared three sets of recombination maps, each of which consisted of five versions. The first set ("stepwise") was 16 Mb long, and the recombination rate was set to one-tenth the mutation rate ($4.6 \times 10^{-10}$) at the central 10 Mb, with a step-wise increase in recombination rate at 3 Mb ends of the chromosome (Fig. 2b), such that the mean recombination rate (after masking 6 Mb of the middle) were 0.1, 0.25, 1, 4, and 10 times the mutation rate. The second set ("narrow high-rec.") was 11 Mb long, and the recombination rate was set to $4.6 \times 10^{-10}$ throughout the chromosome, except a 1 Mb segment in the middle, where recombination rate was elevated such that the mean recombination rate were 0.1, 0.25, 1, 4, and 10 times the mutation rate (Supplementary Fig. S1a). The third set ("uniform") consisted of five uniform recombination maps of 10 Mb with recombination rate of 0.1, 0.25, 1, 4, and 10 times the mutation rate (Supplementary Fig. S1b). In most simulations, we utilized the stepwise maps. We used the narrow high-rec and uniform maps to investigate the effect of different levels of heterogeneity in recombination rate along the genome.

#### Neutral coalescent simulation

To investigate the effect of high-recombining regions on demography inference, we simulated ARGs and mutations with msprime version 1.2.0 (Baumdicker *et al.* 2022) under the standard neutral coalescent with recombination (Hudson 1983). The demography model presented in the main results (demography model 1) consisted of an ancestral population of 1,000,000 diploids splitting into three populations (pop1, pop2, and pop3) at 10,000 generations before the present time. The population size of pop1 was constant at 10,000, and exponential increase and decrease of 10 folds over 10,000 generations were introduced in pop2 and pop3 after the split event. The mutation rate was set to $4.6 \times 10^{-9}$ per generation per site.

To investigate the effect of high-recombining regions on demography inference under demography scenarios with older split, we simulated two other demography scenarios in which the ancestral size is 100,000 (demography model 2) and 10,000 (demography model 3), keeping the split time and fold change in effective population size after the split the same as demography model 1. Because the three scenarios differ only by the ancestral population size, they reflect scenarios with from a recent to moderately old population split event relative to the long-term effective population size.

For each stepwise recombination map (out of five) and demography model (out of three), we simulated 10 replicates of 150 diploid individuals (50 diploids per population). With the narrow high-rec and uniform recombination maps, we simulated one replicate for each map under demography model 1. We recorded the true ARGs in tree sequence format, and also recorded haplotype data in VCF format using tskit version 0.4.1 (Kelleher *et al.* 2018).

#### Forward simulation with background selection

To investigate how background selection impacts on ARG-based demography inference along with the effect of recombination

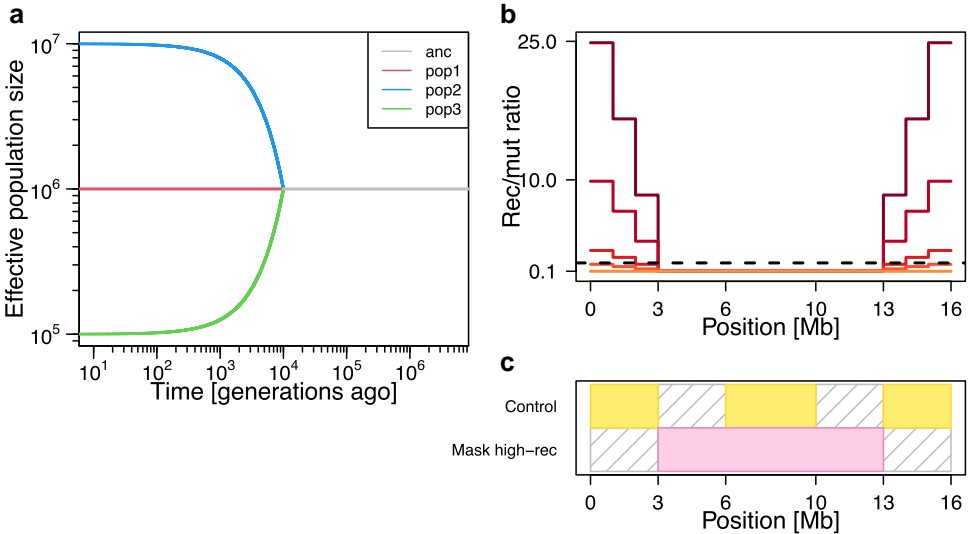

**Fig. 2.** Design of our simulation study. a) Simulated demographic history. We simulated three populations (pop1, pop2, pop3) that split from one ancestral population simultaneously 10,000 generations before the present. After the population split, these three populations follow distinct trajectories of effective population size. b) Recombination maps used in our simulations for a hypothetical chromosome. We considered five scenarios with different recombination maps that are depicted by five solid color-coded lines. In all scenarios, recombination rate (Y axis) is one-tenth of the mutation rate in the middle of the chromosome. Recombination rate increases in a stepwise manner (scenarios with different levels of increase are color-coded from orange to dark red) towards the ends of the chromosome. The horizontal black dotted line indicates the recombination-to-mutation ratio of 1. c) Two settings of demography inference masking different parts of the chromosomes (Colored blocks depict genomic regions used for demography inference. Gray shades depict regions masked from demography inference). We asked whether the presence of high-recombining regions (i.e. towards the chromosome ends in the simulated scenario illustrated in c) in the genome affects demography inference and whether masking them improves demography inference. We applied methods of demography inference on the simulated data using different parts of the chromosome: either including the high-recombining regions ("control," top) or masking the 6 Mb of high-recombining regions (bottom). To control for the total sequence length used in the inferences, we applied a total of 6 Mb masks outside the high-recombining regions in the first setting (top).

landscape, we performed forward simulation using SLiM version 4.0.1 (Haller and Messer 2022), pyslim version 1.0.4 (Haller *et al.* 2019), msprime version 1.2.0 (Baumdicker *et al.* 2022), and tskit version 0.5.5 (Kelleher *et al.* 2018). We only considered demography model 2 (ancestral population size $N_{anc} = 100,000$) to keep the level of rescaling reasonable. We used the five stepwise recombination maps described above and uniformly placed 250 genes with 12 exons of 357 bp with introns of 1,558 bp (equivalent to the genome annotation of the blackcap Bascón-Cardozo *et al.* 2024). We considered five distributions of fitness effect (DFEs) of mutations within coding regions similarly to Johri *et al.* (2021). To model different strengths of background selection, these DFEs differ in the fraction of mutations of four discrete classes of selection coefficient: frequency $f_0$ for class 0 mutations with $s_0 \sim U_{[0, \frac{1}{2N_{anc}}]}$; frequency $f_1$ for class 1 mutations with $s_1 \sim U_{[\frac{1}{2N_{anc}}, \frac{10}{2N_{anc}}]}$; frequency $f_2$ for class 2 mutations with $s_2 \sim U_{[\frac{10}{2N_{anc}}, \frac{100}{2N_{anc}}]}$; and frequency $f_3$ for class 3 mutations with $s_3 \sim U_{[\frac{100}{2N_{anc}}, 1]}$, where $s_i$ is the selection coefficient of mutations of class $i \in \{0, 1, 2, 3\}$ and $U_{[a,b]}$ is a uniform distribution from $a$ to $b$ (Supplementary Table S1). Fitness effect of mutations within exons were modeled to follow the DFE described above, while those in noncoding regions were modeled to be neutral ($s = 0$).

We ran simulation in two steps. In the first step, we simulated only nonneutral mutations (classes 1–3 within coding regions) using SLiM. To implement the model in SLiM, we re-scaled population genetic parameters by 100. We re-scaled the time and population size by 1/100 fold, mutation rate to $100\mu (= 4.6 \times 10^{-8})$ [per bp per generation], and recombination rate to $\frac{1}{2}(1 - (1 - 2r)^{100})$ [per bp per generation]. With this re-scaling, we ran a burn-in of 10,000 generations for the ancestral population and introduced the population split event. One hundred generations after the split (corresponding to 10,000 generations without re-scaling), we recorded the ARG in tree sequence format.

In the second step, using pyslim, msprime, and tskit, we added neutral mutations in coding and noncoding regions on the ARG, and recorded genotypes of samples. Specifically, we first simulated the ancestry of uncoalesced lineages of the ARG ("recapitated" the ARG) using pyslim, with re-scaled population size and recombination map. After sampling 50 diploids per population and simplifying the ARG, we simulated neutral mutations in coding and noncoding regions with re-scaled mutation rate. Finally, we recorded genotype data in VCF format.

### Demography inference

**Masking.** To test whether masking high-recombining regions improves ARG-based demography inference, we applied masks accounting for the length of chromosomes in two ways: masking high-recombining regions or masking the same length of central part of the chromosome without elevated recombination rate ("control"). Specifically, for the step-wise recombination maps (Fig. 2b), we masked 3–6 [Mb] and 13–16 [Mb] regions for control, and 0–3 [Mb] and 13–16 [Mb] regions to mask high-recombining regions (Fig. 2c). For the narrow high-rec. scenario (Fig. S1b), we masked 4.5–5.5 [Mb] to mask the high-recombining region or 10–11 [Mb] for control.

**MSMC2.** For the narrow high-rec. and uniform scenarios, we used four diploid individuals from each population for demography inference with MSMC2 (Malaspinas *et al.* 2016; Wang *et al.* 2020). For the stepwise scenario, we treated 10 simulations as 10 independent chromosomes, and downsampled four diploid individuals (8 haploid sequences) per population without replacement 10 times as 10 "replicates" (Note that they are not true independent replicates because they were sampled from a common ARG for each chromosome). Input multihetsep files were generated from the VCF file and masks for each chromosome

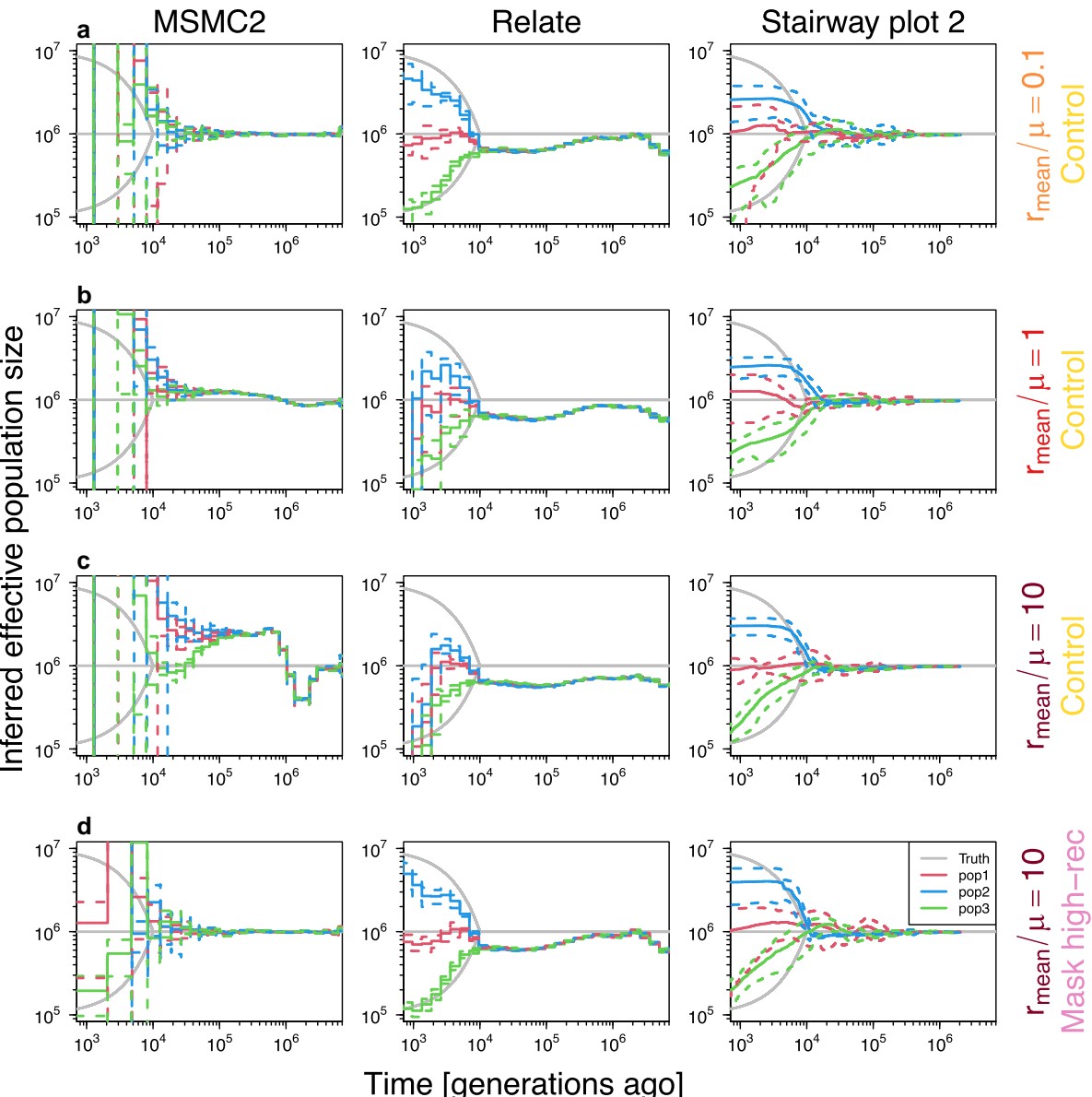

**Fig. 3.** Inference of historical effective population size. The inferences by MSMC2 (left) and Relate (middle) without masking high-recombining regions (a–c) show that ARG-based methods are biased in the presence of high-recombining regions. Removing the high-recombining regions eliminated the bias (d). The results for Stairway plot 2 (right) confirm the expectation that the SFS-based method is less affected by the presence of high-recombining regions. Sample sizes differed among three methods: 4 diploids/population in MSMC2, and 50 diploids/population in Relate and Stairway plot 2. In each panel, gray lines depict the simulated truths (as in Fig. 2a), solid and dashed colored lines depict the mean and mean ± SD of the inferences (see Materials and methods for details on how replicates were treated).

of each downsample of each replicate using genrate_multihetsep.py of msmc-tools (Schiffels and Wang 2020). We ran MSMC2 for each population or population pair to infer historical coalescence rates. The estimates of historical effective population size were obtained as the inverse of the inferred coalescence rate for each population, scaled with the true mutation rate of $4.6 \times 10^{-9}$. The rCCR was obtained by dividing the between-populations coalescence rate with the average within-population coalescence rate. For visualization, we computed the mean and standard deviation of the inferred effective population size and rCCR with a custom script, and the inferred demography older than 1,000 generations ago was plotted (e.g. Figs. 3 and 4).

*Relate.* For the stepwise scenario, we treated 10 simulations as 10 independent replicates. We applied filtering of variable sites

based on the position according to masking conditions using BCFTools version 1.9 (Danecek et al. 2021). We inferred ARGs from the masked VCF using Relate version 1.1.6 (Speidel et al. 2019) specifying the true mutation rate, true recombination maps, and haploid population size of $2N_{anc}$, and inferred demography with two iterations. The estimates of historical effective population size were obtained as the inverse of the inferred coalescence rate for each population, scaled with the true mutation rate of $4.6 \times 10^{-9}$. The rCCR was obtained by dividing the coalescence rate between populations with the average within-population coalescence rate. For visualization, we computed mean and standard deviation of the inferred effective population size and rCCR with a custom script. As for MSMC2, inference of demography older than 1,000 generations ago was plotted (e.g. Figs. 3 and 4).

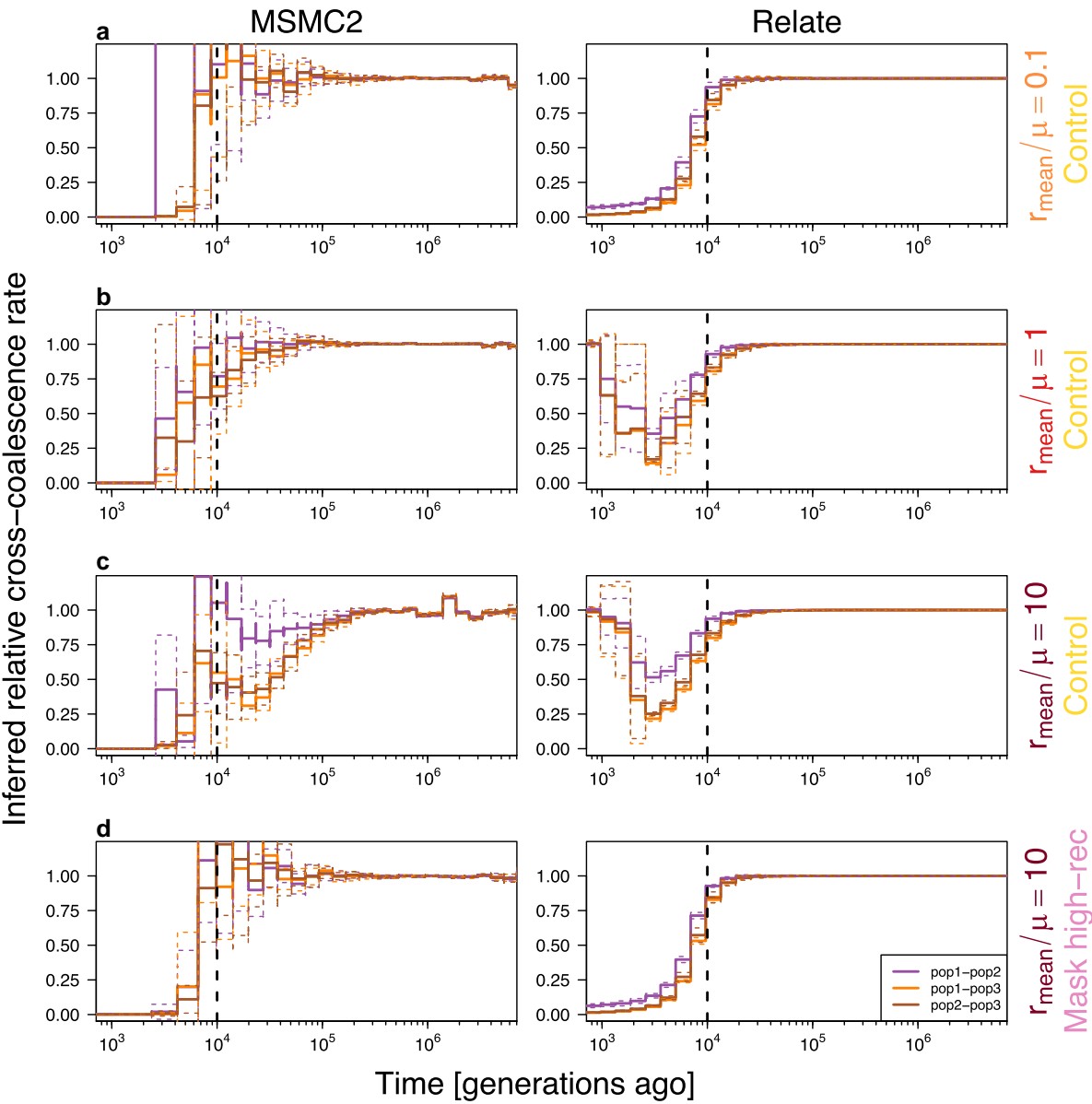

**Fig. 4.** Inference of population split events. The results show that the presence of high-recombining regions biases the inference of population splits (a–c), and removing the high-recombining regions reduces this bias (d). Vertical dotted lines depict the true split time. Colored lines depict inferred rCCR for pairs of populations. Three colors indicate three pairs of populations. Solid lines depict mean of inferences of the 10 down-samples (MSMC2, 4 diploids/ population) or 10 replicates (Relate, 50 diploids/population), and dotted lines depict mean ± SD.

*Stairway plot 2.* We ran Stairway plot 2 version 2.1 (Liu and Fu 2020) for the stepwise scenario. We treated 10 simulations as 10 independent replicates. We split the VCF by population applying masks with VCFTools version 0.1.16 (Danecek *et al.* 2011). We computed the unfolded SFS and prepared blueprint configuration files using custom scripts, and ran Stairway plot 2 with default parameter values. For visualization in Fig. 3, we computed mean and standard deviation of the inferred effective population size with a custom script. As for MSMC2 and Relate, inference of demography older than 1,000 generations ago was plotted (Fig. 3).

### Coalescence time analysis

*MSMC2.* We focused on two haploids of the first chromosome (simulation run) of the first downsample in pop1, and compared true TMRCA recorded in the true ARG (in TreeSeq format) and inference by MSMC2. We extracted the true TMRCA of the focal pair

of haploid genomes in TreeSeq with tskit. To obtain inference by MSMC2, we ran the decode program of MSMC2 with decode -m 0.0092 -r 0.00736 -I 0,1 -t 32 -s 1000. Based on the output of decode, we recorded the index of epoch with the highest probability for each window. We aligned true and inferred TMRCA treating an intersected range as a unit, and computed Spearman's correlation coefficient in R version 4.3.1 (R Core Team 2023).

*Relate.* We focused on TMRCA of the entire genealogy of 300 haploid genomes of the first simulation replicate. We extracted TMRCA along the chromosome from the true ARG in TreeSeq using tskit. To obtain TMRCA along the chromosome of the ARG inferred by Relate, we converted the genealogies (in mut and anc format) to TreeSeq using RelateFileFormats program in Relate, and extracted TMRCA along the chromosome using

tskit. We aligned true and inferred TMRCA treating an intersected range as a unit, and computed Spearman's correlation coefficient in R.

### Recombination inference

To evaluate whether LD-based methods of recombination map inference can be used to define high-recombining regions to mask for ARG-based demography inference, we inferred recombination maps using LDhat (Auton *et al.* 2013) and pyrho (Kamm *et al.* 2016; Spence and Song 2019) for each population using different sample sizes ($n \in \{5, 10, 20\}$) based on one replicate of simulated data under the demography model 1 with the stepwise recombination maps. For both LDhat and pyrho, we analyzed the genetic variation data of one simulation replicate.

*LDhat.* We created a lookup table for $n = 20$ (40 haploids) using the complete program of LDhat with `-n 40 -rhomax 100 -n_pts 101 -theta 0.016`, where $\theta = 0.016$ was based on the Watterson's estimator applied to the simulated data. Then, we created lookup tables for $n = 5$ and $n = 10$ based on the lookup table of $n = 40$ using the lkgen program of LDhat. For inference of recombination rate, we first split VCF into 2 kb sliding windows overlapping by 1 kb using BCFTools, and we converted each chunk into LDhat format using VCFTools with the `--ldhat` flag. To estimate local recombination rates, we ran the interval program of LDhat with `-samp 5,000 -bpen 5 -its 10,00,000` for each chunk. We summarized the inference using the stat program of LDhat, concatenated chunks (computing the mean values for overlapping intervals), and computed mean population recombination rates in nonoverlapping 10-kb sliding windows using a custom script. We computed sliding-window $\rho/\theta$ by dividing the estimated map of $\rho = 4N_e r$ by genome-wide estimation of $\theta$ based on the Wattersons's estimator.

*pyrho.* We created lookup tables for $n \in \{5, 10, 20\}$ using pyrho make_table specifying the true demography for each population. We estimated recombination rates using pyrho optimize with `--windowsize 50 --bpen 75`, the parameter values of which were determined using pyrho hyperparam. We computed mean recombination rate in 10-kb sliding windows using a custom script. We computed sliding-window $r/\mu$ by dividing the estimated map of $r$ by the true mutation rate $\mu = 4.6 \times 10^{-9}$.

*iSMC.* To investigate whether joint inference of local recombination and mutation rates can identify regions to mask with recombination rate much higher than mutation rate, we ran iSMC version 0.0.25 (Barroso *et al.* 2019; Barroso and Dutheil 2023) for each population for one replicate of simulated data under the demography model 1 with the stepwise recombination maps. Using one diploid per population, we ran iSMC with four classes of $\rho$, four classes of $\theta$, and one class of $\tau$. Based on the decoded recombination and mutation maps, we computed the genomic landscape of $\rho/\theta$, and checked whether high-recombining regions have inferred $\rho/\theta > 1$.

## Empirical study

### Data

We used phased whole-genome resequencing (WGR) data of 179 blackcaps (Ishigohoka *et al.* 2023, NCBI BioProject PRJEB66075), and unphased WGS data of five garden warblers and three African hill babblers (Delmore *et al.* 2020). We computed mean recombination rate in 10-kb sliding windows along the blackcap genome based on Bascón-Cardozo *et al.* (2024).

### Demography inference

*MSMC2.* We first applied callability masks to the blackcap genome and defined respective high- and low-recombining halves of the genome for each population or population pair. Specifically, we chose for each population at most four individuals with mean read depth of at least 15×, excluding pairs of related individuals based on kinship coefficient (Manichaikul *et al.* 2010) computed using relatedness2 option in VCFTools. We created mask file per individual using bamCaller.py of msmc-tools (Schiffels and Wang 2020) and merged them for each population or population pair using bedtools merge (Quinlan and Hall 2010). The mask for each population or population pair was applied on the blackcap recombination map (Bascón-Cardozo *et al.* 2024), and we ordered genomic intervals within the unmasked regions according to the recombination rate. Regions in the first and the second halves were defined as the low- and high-recombining halves.

After defining the regions to be used for inference, input multihetsep files were generated from the phased VCF and the mask file using genrate_multihetsep.py of msmc-tools (Schiffels and Wang 2020). We ran MSMC2 for each population or population pair to infer historical coalescence rates. The estimates of historical effective population size were obtained as the inverse of the inferred coalescence rate for each population, scaled with a mutation rate of $4.6 \times 10^{-9}$ estimated in the collared flycatcher (Smeds *et al.* 2016). The rCCR was obtained by dividing the between-population coalescence rate with the average within-population coalescence rate.

*Relate.* Relate requires haplotype data with polarized mutations. We polarized biallelic SNPs in blackcaps using allele frequencies in two outgroup species, garden warblers ($n = 5$) and African hill babblers ($n = 3$). Specifically, after removing SNPs with more than two alleles including the three species, we split blackcap SNPs into the following five categories.

1) Sites at which all garden warblers had missing genotype.
2) Sites fixed in garden warblers
3) Sites segregated among garden warblers and missing in all African hill babblers
4) Sites segregated among both garden warblers and African hill babblers
5) Sites segregated among garden warblers and fixed in African hill babblers

For these categories, we applied the following heuristics to polarize mutations. For sites of the first type, we defined the minor allele among blackcaps to be the derived state (i.e. the major allele is the ancestral state). For sites of the second type, we defined the allele possessed by garden warbler to be the ancestral state. For sites of the third or fourth type, we defined the minor allele among blackcaps to be the derived state (i.e. the major allele is the ancestral state). For sites of the fifth type, we defined the allele possessed by African hill babbler to be the ancestral state.

We defined low- and high-recombining halves of the genome in BED format based on the blackcap recombination map (Bascón-Cardozo *et al.* 2024). Based on these BED files to mask high/low-recombining half and repeats retrieved from UCSC Genome Browser tracks (Raney *et al.* 2023) for the blackcap assembly (GenBank: GCA_009819655.1), we made a mask file in FASTA format for each condition using BEDTools maskfasta. Using the phased and polarized VCF, recombination map, and the mask, we ran Relate to infer genealogies with mutation rate of $4.6 \times 10^{-9}$

and effective population size of 500,000. We inferred demography from the genealogies using `RelateCoalescenceRate` program of `Relate` with mutation rate of $4.6 \times 10^{-9}$ and five times of iterations. The estimates of historical effective population size were obtained as the inverse of the inferred coalescence rate for each population with rescaling of time by generation time of 2 years (Delmore *et al.* 2020). The rCCR was obtained by dividing the between-population coalescence rate with the average within-population coalescence rate.

## Results

### Effect of wide high-recombining regions on inference of effective population size

We first investigated the effect of high-recombining regions on the inference of historical effective population size under the demography model 1 ($N_{anc} = 1,000,000$ with a recent population split). Under the control condition (i.e. without masking high-recombining regions), we estimated demography using two ARG-based methods, MSMC2 and `Relate`, and one method based on SFS, `Stairway plot 2` (Fig. 3). Inference by MSMC2 was too noisy in the recent past to inspect the timing of the split event based on inferred effective population size especially in scenarios with mild or no elevation of recombination rates at the chromosomal ends (Fig. 3, left column. Results of all five recombination landscapes are in Supplementary Fig. S2). This noisiness in MSMC2 inference is unlikely to be the effect of recombination rate, as the inference was noisy even for the scenario without a high-recombining region (Fig. 3a, left column), but rather results from insufficient data (8 haploids of 10 chromosomes of 16 Mb) and the simulated split time being too recent compared to the effective population size. Indeed, in demography models 2 and 3 ($N_{anc} = 100,000$ and $N_{anc} = 10,000$, respectively, representing scenarios with incrementally older population splits), the inference in the recent past was less noisy (Supplementary Figs. S3 and S4). When the mean recombination rate was lower than or equal to the mutation rate, MSMC2 inferred effective population size accurately in the deep past (Fig. 3a, b, left column). However, when the mean recombination rate was greater than the mutation rate, the inference of effective population size by MSMC2 was systematically deviated in the deep past with a characteristic wave-shaped pattern in the skyline plot (Fig. 3c, left column). Inference by `Relate`, which exploited a total of 300 haploids from the three subpopulations, was less noisy compared to MSMC2 even in the recent past (Fig. 3, middle column. Results of all five recombination landscapes are in Supplementary Fig. S2). This reflects the high scalability of `Relate`, because the inference by `Relate` also became noisier in the recent past when a comparable sample size to MSMC2 was used (Supplementary Figs. S8 and S9). In the case without high-recombining regions (Fig. 3b, middle column), the apparent split time as well as the post-split trajectories of the effective population sizes were accurately inferred, albeit slightly smaller effective population size before the split event was estimated. However, as the mean recombination rate increased, the effective population sizes of pop1 (red, constant size) and pop2 (blue, exponential growth) were systematically underestimated after the population split. We observed consistent effects of high-recombining regions on ARG-based demography inference under demography models 2 and 3 (Supplementary Figs. S3 and S4). In contrast to the two ARG-based methods, demography inference by SFS-based `Stairway plot 2` was robust to the presence of high-recombining regions (Fig. 3, right column).

To investigate whether the deviations in the demography inferences by ARG-based methods are due to errors in inferred local ARGs within the high-recombining regions or global errors throughout the entire chromosome, we compared coalescence time metrics in inferred (representation of) ARGs with the simulated truth (detailed in Materials and methods). In both ARG-based methods, correlation between the inference and the truth was reduced specifically in the high-recombining regions (Supplementary Figs. S10 and S11). Furthermore, masking high-recombining regions improved demography inference in both methods (Fig. 3d, Supplementary Figs. S2, S3, and S4). Our findings indicate that ARG-based demography inference is affected by localized errors within the high-recombining regions.

### Effect of wide high-recombining regions on the inference of population splits

Next, we investigated the effect of high-recombining regions on ARG-based inference of population split events under the demography model 1 by inferring historical relative cross-coalescence rate (rCCR). RCCR is a metric informative of split events and gene flow between a pair of populations: it increases from 0 to 1 backwards in time (i.e. it drops from 1 to 0 forward in time) at the population split event, and the rate of this change in respect to time essentially depicts how fast the split occurred (Schiffels and Durbin 2014). For recombination landscapes with high-recombining regions, inferred rCCR based on MSMC2 started to drop (forward in time) earlier than the true split time (Fig. 4. Results for all recombination maps are in Supplementary Fig. S5.). This was especially true for subpopulation pairs involving pop3 (exponential reduction in effective population size) as the mean recombination rate increased. This pattern is consistent with the spurious older split based on effective population size inferred by MSMC2 with high-recombining regions (Fig. 3c, left column). In inferences by `Relate`, rCCR decreased (forward in time) at the true split time, but increased again towards the present time in scenarios with high-recombining regions (Fig. 4. Results for all recombination maps are in Supplementary Fig. S5.) These impacts by high-recombining regions on inference of rCCR was consistent in demography models 2 and 3 with smaller ancestral population size (i.e. older population split relative to ancestral population size) (Supplementary Figs. S6 and S7). Importantly, we were able to mitigate these effects by masking the high-recombining regions (Fig. 4d). To summarize, our simulation study illustrates that the presence of high-recombining regions affects ARG-based inference of both historical effective population size and population split time. We showed that these effects can be mitigated by masking high-recombining regions of the genome in the analysis.

### Effect of a narrow high-recombining region on demography inference

To further explore how different levels of heterogeneity in the local recombination rate along chromosomes might affect ARG-based demography inference, we simulated the demography model 1 with two additional sets of recombination maps, namely "uniform" and "narrow high-rec." (Supplementary Fig. S1; Materials and methods for details). For the uniform scenario, inference by both MSMC2 and `Relate` were affected by recombination rate similarly to the stepwise recombination maps as the recombination rate increased (Supplementary Fig. S12). For the narrow high-rec. scenario, inference by MSMC2 was slightly affected when the mean recombination rate was greater than the mutation rate (Supplementary Fig. S13). In contrast, inference

by `Relate` was robust to the presence of the narrow high-recombining region. These results and the demography inference by `Relate` with small sample size (Supplementary Figs. S8 and S9) indicate that the ARG-based methods are affected by a wide coverage of regions with recombination rate greater than mutation rate, especially when the sample size is small.

## Inference of recombination maps to define high-recombining regions

Although we have shown that masking high-recombining regions based on a true recombination map alleviates the impact of high-recombining regions on ARG-based demography inference, we need to use estimated recombination maps in practice. To investigate whether recombination maps inferred by LD- and SMC-based methods can be used to define high-recombining regions to mask for ARG-based demography inference in practice, we compared inferred recombination maps based on `LDhat`, `pyrho`, and `iSMC` with the truth. Despite the high correlation between the inference and the truth (Supplementary Tables S2, S3, and S4), all `LDhat`, `pyrho`, and `iSMC` underestimated recombination rate within high-recombining regions (Supplementary Figs. S15, S16, and S17). Nonetheless, the inferred recombination rate within high-recombining regions was higher than mutation rate, especially in cases with the most serious impacts on ARG-based demography inference (mean recombination rate of 4 and 10 times as high as the mutation rate), indicating the practical possibility of applying a threshold of inferred $r \geq \mu$ to define high-recombining regions.

## Effect of background selection

Demography inference is known to be affected by background selection (Johri *et al.* 2021, 2022). Although high-recombining regions impact ARG-based demography inference, they may buffer the effect of background selection on demography inference, because neutral mutations are less affected by nearby mutations under purifying selection if recombination is frequent (Hudson and Kaplan 1995; Nordborg *et al.* 1996). To investigate how background selection affect ARG-based demography inference with different recombination landscapes, we simulated demography model 2 with the stepwise recombination maps with different levels of background selection (Materials and methods for details). In contrary to our expectation, background selection impacted demography inference constantly for all recombination maps. Background selection affected the scaling of inferred historical effective population size (Supplementary Figs. S18 and S22) and time (Supplementary Figs. S20 and S24) irrespective of high-recombining regions being present or masked (Supplementary Figs. S19, S21, S23, and S25).

## Empirical analysis

Birds lack PRDM9 and their recombination is characterized by higher rates and wider hotspots compared to PRDM9-dependent recombination in primates (see Introduction). Additionally, per-generation mutation rate is lower in many birds than in humans (Bergeron *et al.* 2023). These factors are manifested as wider breadth of genomic regions with high recombination-mutation ratio, which could bias ARG-based methods for demography inference as shown in our simulation study. To address the relevance of our findings to empirical applications, we revisited genome data of the Eurasian blackcap as a representative of birds, which lack PRDM9. To investigate the effect of recombination rate on ARG-based methods of demography inference with the blackcap dataset, we split the blackcap genome into low- and high-recombining halves and compared demography inference by

`MSMC2` and `Relate` between these two halves (see Materials and methods for details). Inference by `MSMC2` showed apparent effects of high-recombining regions consistent with our simulation. Historical effective population size inferred using the high-recombining half had characteristic wave-shaped trajectory in the deep past of the skyline plot (Fig. 5b) compared to that using the low-recombining half (Fig. 5a). The apparent split time between populations based on effective population size was older using the high-recombining half (Fig. 5b) than using the low-recombining half (Fig. 5a). In line with this, direct comparison of inferred rCCR for pairs of blackcap populations between the high- and low-recombining halves revealed inference of systematically older split time using the high-recombining half than the low-recombining half (Fig. 5c, d). These differences are consistent with our simulation study (Figs. 3 and 4), indicating that the effect of high-recombining regions on inference by `MSMC2` is relevant for empirical analysis. In contrast, `Relate` was more robust to the difference in recombination rate between the two conditions (Supplementary Fig. S14). This difference between `MSMC2` and `Relate` is in line with our simulation i.e. better inference with larger sample size, enabled in `Relate` due to its high scalability.

## Discussion

Our results suggest that demography inference using ARG-based methods should be carried out with caution in organisms that are likely to harbor recombination landscapes distinct from humans, which were initially utilized to validate these methods. In many animals with functional PRDM9, including humans, recombination events are concentrated in narrow recombination hotspots (Myers *et al.* 2010; Stevison *et al.* 2016). Therefore, we expect that ARG-based methods will be robust in these organisms. In contrast, other species have high-recombining regions more widely distributed around genomic features along the genome (Singhal *et al.* 2015; Baker *et al.* 2017; Kawakami *et al.* 2017) and thus ARG-based methods can be more susceptible to the effect of high-recombining regions. Although inference of fine-scale recombination maps can be inaccurate especially when the recombination rate is higher than the mutation rate (Spence and Song 2019; Raynaud *et al.* 2023), we demonstrated that masking high-recombining regions, which may practically be defined based on population genomic inference of recombination maps as regions with $\rho/\theta \geq 1$ or $r/\mu \geq 1$, improves the ARG-based demography inference in such cases. We additionally showed that background selection impacts ARG-based demography inference on top of the effects of recombination landscape. Because the efficacy of weak selection depends on effective population size (Ohta 1973), the relative impacts between the recombination landscape and background selection may differ between organisms with different long-term effective population size.

In practice, chromosomes can co-vary in size, genomic architecture, and recombination rate (Kawakami *et al.* 2014; Singhal *et al.* 2015; Martin *et al.* 2019; Bascón-Cardozo *et al.* 2024), potentially leading to differences among chromosomes with respect to applicability of ARG-based methods. In addition to high-recombining regions, masking additional genomic features may be necessary for demography inference if large blocks that do not represent neutral evolution exist in the genome. For example, large polymorphic inversions under long-term balancing selection (Küpper *et al.* 2015; Lamichhaney *et al.* 2015; Knief *et al.* 2016, 2017; Kim *et al.* 2017; Mérot *et al.* 2021; Giraldo-Deck *et al.* 2022; Hager *et al.* 2022; Harringmeyer and Hoekstra 2022) may be excluded, which may

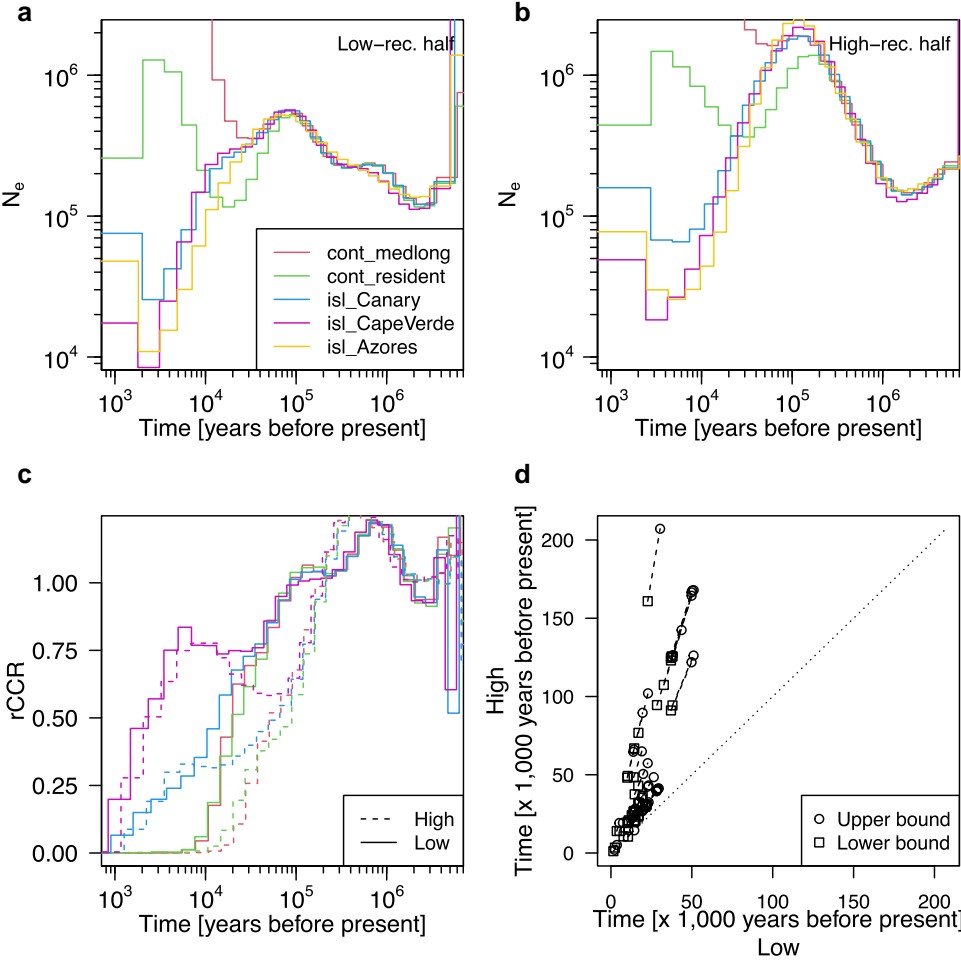

**Fig. 5.** High-recombining regions can affect demography inference in empirical analysis. a), b). Inference of historical effective population size by MSMC2. Results for five exemplified blackcap populations are shown using the lower a) or the higher b) half of the genome based on local recombination rates. c) Inference of relative cross-coalescence rate (rCCR) with MSMC2 between Azores population and each of all other four populations in a) and b) using the lower (solid lines) and the higher (dotted lines) half of the genome based on local recombination rates. d) Comparison of split times inferred by MSMC2 using the lower higher halves of the genome based on recombination rates. Segments represent inference between 45 pairs of 10 populations. Two ends of a segment represent the lower and upper boundaries of two consecutive discretized epochs between which rCCR crosses the threshold of 0.5.

leave little data for inference in species with small genomes. We suggest to run simulations tailored to the focal study species to assess whether ARG-based methods can be used with sufficient confidence. This is especially necessary in species without functional PRDM9, with broad high-recombining regions, high genome-wide mean recombination rates, small genomes, highly heterogeneous chromosomes, and large structural variations.

An SFS-based method for demography inference, `Stairway plot 2`, performed well without high-recombining regions and even better with high-recombining regions in our simulation study. We propose that this accuracy under the presence of high-recombining regions can represent a general characteristic of SFS-based methods that they benefit from high-recombining regions, from which ARG-based methods suffer. The problem of high-recombining regions for ARG-based methods is the fact that branches of genealogies are not represented by mutations (Hayman *et al.* 2023). In other words from the perspective of mutations, ARG-based methods suffer from independence of mutations in a local genomic window. This independence of mutations, however, is the assumption to compute SFS (Gutenkunst *et al.* 2009), allowing SFS-based methods to perform accurately with high-recombining regions. Localized errors in the inference of (representation of) genealogies within high-recombining regions in our study indicate that the issue of high-recombining

regions in ARG-based methods is not specific to demography inference but can be critical in other applications, including inference of selection (Speidel *et al.* 2019; Stern *et al.* 2019; Hejase *et al.* 2020). In regions with high recombination rates, inferred genealogies may be too inaccurate to perform ARG-based selection tests, while SFS-based methods may be used on the local variation data (Tajima 1989; Fay and Wu 2000). Combining ARG- and SFS-based approaches, giving them complementary weights according to the local recombination rate, may make the most of the variation data in demography inference.

In this study, we demonstrated that the recombination landscape can influence ARG-based approaches of population genomics. Although the true ARGs should have rich information on the population history and evolutionary processes, the effects of errors in inferred local ARGs within regions of elevated recombination rate are, in some cases, not negligible. Our findings are likely relevant not only to birds but in a wide range of species, because PRDM9 has been lost at least thirteen times independently in vertebrates (five clades of ray-finned fish, four clades of amphibians, a clade of lizards, the entire clade of birds and crocodiles, and two clades of mammals (dogs and platypus) Cavassim *et al.* 2022). In addition to the recombination rate, other factors, such as the genomic landscape and spectra of mutation

(Wu *et al.* 2020; Jiang *et al.* 2021; Monroe *et al.* 2022; Sasani *et al.* 2022), local effective population size (reflecting selection: Nielsen 2005; Ellegren and Galtier 2016; Burri 2017), and effective migration rate (reflecting barriers to gene flow: Westram *et al.* 2022) are distributed nonuniformly along the genome, and they may similarly affect population genomics summary statistics and inferences. Novel approaches jointly modeling heterogeneity of some of these factors are emerging (Barroso and Dutheil 2023; Laetsch *et al.* 2023; Korfmann *et al.* 2024). Nonetheless, we highlight that evaluating the performance and limitation of population genomics methods under noncanonical parameter space relevant in individual cases is necessary to draw meaningful interpretations.

## Data availability

Scripts used for the simulations and input data, processed output data, and scripts for the empirical analyses are found in Zenodo (https://doi.org/10.5281/zenodo.14590030).

Supplemental material available at GENETICS online.

## Acknowledgments

We thank Julien Dutheil, Linda Odenthal-Hesse, and Diethard Tautz for feedback.

## Funding

M.L. was supported by the Max Planck Society (Max Planck Research Group grant MFFALIMN0001) and the DFG [project Nav05 within SFB 1372 – Magnetoreception and Navigation in Vertebrates (395940726)]. J.I. was funded by the International Max Planck Research School for Evolutionary Biology.

## Conflicts of interest

The author(s) declare no conflicts of interest.

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

*Editor: G. Coop*