## [Peer Review File · Genetics]

High-recombining genomic regions affect demography inference based on ancestral recombination graphs

Jun Ishigohoka and Miriam Liedvogel

NOTE: The reviews and decision letters are unedited and appear as submitted by the reviewers.

In extremely rare instances and as determined by a Senior Editor or the EIC, portions of a review may be redacted. If a review is signed, the reviewer has agreed to no longer remain anonymous.

The review history appears in chronological order.

Review Timeline:

Submission Date:	2024-02-05
Editorial Decision:	2024-04-08
Resubmission Received:	2024-12-09
Accepted:	2024-12-23

April 7, 2024

GENETICS-2024-306830

High-recombining genomic regions affect demography inference

Dear Dr. Ishigohoka:

Three experts in the field have reviewed your manuscript, and I have read it as well. The reviewers appreciated the work highlighting the issue of haplotype-based methods in high recombination regions. While your manuscript is not currently acceptable for publication in GENETICS, we would welcome a substantially revised manuscript. Both reviewers have comments and concerns to be addressed in a revised manuscript. You can read their reviews at the end of this email.

The reviewers have a number of good recommendations about the simulations. While you do not have to run all these additional simulation, however, you should provide a detailed comments in response. In particular you should justify your choice of parameters more, and likely run simulations with more ancient population size change. We look forward to receiving your revised manuscript. Please let the editorial office know approximately how long you expect to need for revisions.

Upon resubmission, please include:

1. A clean version of your manuscript;
2. A marked version of your manuscript in which you highlight significant revisions carried out in response to the major points raised by the editor/reviewers (track changes is acceptable if preferred);
3. A detailed response to the editor's/reviewers' feedback and to the concerns listed above. Please reference line numbers in this response to aid the editor and reviewers.

Your paper will likely be sent back out for review.

Additionally, please ensure that your resubmission is formatted for GENETICS
<https://academic.oup.com/genetics/pages/general-instructions>

Follow this link to submit the revised manuscript: Link Not Available

Sincerely,

Graham Coop
Associate Editor
GENETICS

Approved by:
David Begun
Senior Editor
GENETICS

Reviewer #1 (Comments for the Authors (Required)):

The manuscript submitted by Ishigohoka and Liedvogel with their evaluation of methods that approximate the ARG with the SMC for demography inference is somehow relevant and might appeal to some Genetics readers. I enjoyed reading this manuscript, especially the guidelines on how to perform demography inference using methods based on ARG in organisms with high recombination rates relative to mutation rates.

But, given the scope of their investigation, as it is presented now, it adds marginally to what was done before in terms of theory and method evaluation of ARG-based methods. Lewanski et al. (2024) review on ARG addressed some problems inherent in the ARG inference framework, especially the one pointed out by the authors (when $\rho > \theta$). It is a problem of identifiability: high rates of recombination relative to the rates of mutation, make the underlying ARG unidentifiable (Hayman et al. 2023), consequently, $\rho > \theta$ affect any parameter computation (TMRCA and N_e). Soni et al. (2024) addressed the rate variability of mutation and recombination rates on the inference of the demographic model and parameters and DFE from the site-frequency spectrum (which seems to be more robust to $\rho > \theta$ as accessed by the present manuscript). We already know that linked selection messes up everything, especially in the form of background selection (Johri et al. 2021), causing the identification of the wrong demographic model and the wrong inference of DFE. Rate variability makes things even

worse. Solutions to use recombination (and mutation maps) when developing the null model for inference have already been proposed by Johri et al. (2021, 2022), in the context of rate variability and background selection.

While their simulations as it is now add a bit to our understanding of the impact of high recombination rate on demographic inference, I think their recommendations to empiricists are relevant and somehow lacking in the literature. High recombination rates are a problem, but if you have the right resources you can alleviate it. It is important to show with simulations and analysis of data how to do it properly. It might be a good idea to show how to deal with variable recombination rates, for example, when having one good assembly. With this, we can have an estimation of the recombination map (for example, using Barroso et al. 2019), and use it to remove SNPs in high recombining regions before performing the demographic inference. I think this manuscript would attract empiricists if expand on a little bit on the recommendation side, for example, showing how errors in the inference of recombination maps might affect the inference.

Given this things, I am not sure if Genetics would be an appropriate place for this manuscript as it is presented now. I think there are two main ways that the authors could follow them to make their work more impactful:

1) make the work more general on how higher rates of recombination relative to the mutation rates impact the inference of demography with SMC-based methods; here we should see efforts to make the simulations easy to tease apart each confounding evolutionary force (see more details on the major reviews): start with models of simple demography (to access the impact on the inference of population size changes). Incorporate more complex structures, like gene flow or population size changes, but make sure to isolate the impact of each. I would also suggest the incorporation of background selection, although I am aware of the amount of work if it would be the case: learning new things and implementing them. But there are many resources out there to help with this. It would be cool to see the results for when you have islands of high recombination and background selection (see more on this below). If some of those things were accomplished, it would appeal to a broader audience of Genetics;

2) If the authors preferred to keep the narrative of non-model organisms, I think more work should be done on the recommendation part. Indeed, the species that the authors presented empirical results is not a model organism, but they had information to use to mitigate the problem with high rates of recombination that most empiricists don't have. What would be then the recommendations for them? If those things could be incorporated, I think it would appeal to an audience of a more ecological Journal, like Molecular Ecology Resources, for example.

****Major reviews****

First, I think the work might benefit if it started showing results for simulations of simpler demographic scenarios to assess only how high recombination rates might affect the inference. It can be done for example, by simulating only one population with constant size with genomic islands of $\rho/\theta \ll 1$ spaced by regions with where $\rho/\theta = 1$. You also could simulate a genomic with a constant r where $\rho/\theta \ll 1$ (or you can sample many random values of r from a uniform distribution, making it vary quite a lot throughout the genome). Having set the baseline for comparison, you can start making the demographic models more complex, for example with 2-epochs with sudden population size increase/decrease. Finally, compare models where you include exponential size changes with varying (high) recombination rates. For the more complex demographic models, I suggest simulating a control where the $\rho/\theta = 1$.

I think the manuscript needs a better organization, especially the methods and results parts, which seem to overlap and diverge quite a lot. I found many sentences in the results that should be in the methods, for example, if the methods were sufficient, you don't need to write again what you did in the results part.

Finally, I think you should get rid of the "non" model organisms argument from your manuscript. Although it is somehow true that model assumptions for SMC methods are around the idea of $\rho/\theta = 1$, which is, coincidentally, the case for humans, there is nothing in the assumption specific to a model organism. I suggest instead talking about organisms or genomes with elevated recombination rates relative to the mutation rate (it might be that the genome, in general, is like this or some islands are like this).

It would be nice to see (and maybe, more important to the genetics readers) how genomic regions with $\rho/\theta \ll 1$ interacting with background selection affect the demographic inference with SMC-based methods. I know some of these methods are recommended to use a set of neutral sites. But for organisms where the genome is rich in functional content, this is almost impossible. We know that BGS misleads the inference to population growth, but the investigation where done around with simulation $\rho/\theta = 1$ (Johri et al. 2021). I am wondering what high local recombination rates would do to regions where selection is strong. As pointed out by Cousins et al. (2024), when the strength of BGS increases, the distribution of TMRCA gets younger. Perhaps high recombination rates would decouple mutation from the linked selection? I know it would include more work, but with SLiM it is easy to set up simulations like these.

****Minor reviews****

- Lines 1-2: remove. I don't see the necessity to direct the abstract to non-model organisms. Model organisms also have regions of high recombination relative to mutation rate (see Drosophila for example). You can remove it and make it appeal to a broader

audience. You have some interesting findings that apply to demography inference broadly.

- Lines 10-11: Replace "non-model organisms" for species.

- Line 21: Remove "reference". A reference genome is not a real entity, and here you are talking about a feature of a species genome.

- Lines 23-24: "This part isn't that accurate: "Variation in genomes allows inference of historical demographic parameters". Sometimes it is true in some context (when enough data is available) but most of the time it isn't. Perhaps remove this part and connect the first idea of this paragraph with the next paragraph. "Population history affects the patterns of genetic variation which can be inferred with population genomic methods. Many approaches were developed to take advantage of whole-genome data ..."

- Lines 24-37: Might be removed to sell this manuscript to a broader audience. Keep the first phrase as I pointed out in the comment above and go to the next paragraph.

- Lines 35-37: This is interesting but I don't think it should come first in your introduction. Maybe the next paragraph has more interesting information and is directly linked with your question, which as I understand, is to benchmark SMC methods when recombination rate (r) > mutation rate (μ) ($r > \mu$).

- Line 41: change chromosome to region (because are recombining regions or segments within chromosomes).

****Citations****

Barroso, G. V., Puzović, N. & Duthheil, J. Y. Inference of recombination maps from a single pair of genomes and its application to ancient samples. *_Plos Genet_* ****15****, e1008449 (2019).

Cousins, T., Tabin, D., Patterson, N., Reich, D. & Durvasula, A. Accurate inference of population history in the presence of background selection. *_bioRxiv_* 2024.01.18.576291 (2024) doi:10.1101/2024.01.18.576291.

Hayman, E., Ignatieva, A., & Hein, J. (2023). Recoverability of ancestral recombination graph topologies. *Theoretical Population Biology*, 154, 27-39. <https://doi.org/10.1016/j.tpb.2023.07.004>

Johri, P. *_et al._* The impact of purifying and background selection on the inference of population history: problems and prospects. *_Mol Biol Evol_* (2021) doi:10.1093/molbev/msab050.

Johri, P. *_et al._* Recommendations for improving statistical inference in population genomics. *_Plos Biol_* ****20****, e3001669 (2022).

Lewanski, A. L., Grundler, M. C. & Bradburd, G. S. The era of the ARG: An introduction to ancestral recombination graphs and their significance in empirical evolutionary genomics. *_PLOS Genet._* ****20****, e1011110 (2024).

Reviewer #2 (Comments for the Authors (Required)):

The paper "High-recombining genomic regions affect demography inference" by Ishigohoka and Liedvogel presents a study of how specific methods for demographic inference perform under high recombining regions.

Many modern methods to infer the history of population sizes from genetic data use haplotype information in some way or another. This includes MSMC2 and Relate, the two studies compared here. The problem this paper addresses is that high recombining regions somewhat break the assumptions of these methods, in that they try to infer local haplotype structure to help reconstructing (parts of) the ARG, which may become impossible under high recombination.

The authors show that indeed high-recombining regions pose a problem for these methods, while another method that relies purely on the site frequency spectrum, ignoring phase and haplotypes, is - unsurprisingly - not affected by recombination rates.

I think the paper is very clear and well written overall.

I only have two main comments:

1.) I think the comparison between MSMC2 and Relate is made difficult by the fact that MSMC2 uses only 8 chromosomes, while Relate uses 300 (page 7). The reason for this is of course simply the different limits of the methods, but I wonder whether one should add a fourth column to Figure 2 and a third column to Figure 3, with Relate run on only 8 chromosomes? That way, it would become clear whether the noisiness is due to insufficient data (as the authors suggest on page 7) or due to some inherent approximations made by the two methods. At the same time it would leave Relate's superior performance on 300

haplotypes apparent to the reader. Independent on whether the authors agree to add this analysis, I would suggest to add the number of analysed haplotypes somehow to the title line above the plots in Figures 3 and 4, so something like (MSMC2 (8 hapl.), Relate (300 hapl.)), to make clear to the reader that what they are looking at is hard to compare.

2.) Related to the same point, and perhaps an alternative way to improve comparability: I wonder why the authors chose a demographic history with such a recent focus. Given that MSMC/PSMC mostly cover times much older than that (with the parameters considered here more like 10^5 rather than 10^4 generations ago) it would be quite interesting to better understand the performance under high recombination for population size changes in the deeper past. An example could be one with periodic growths and declines (https://popsim-consortium.github.io/stdpopsim-docs/stable/catalog.html#sec_catalog_homsap_models_zigzag_1s14). This would then also make the comparison of the two methods - even with drastically different amounts of data input - more useful, as they both then operate in their intended parameter range.

... and a few more minor comments:

3.) In Figure 1, the schematic, I noticed that the example trees change a bit more drastically from one recombination event to the next than expected. In particular in B, it appears that trees left and right of a recombination break point differ in multiple internal branch lengths. I think this shouldn't be possible: A single recombination event can break an internal branch and "re-glue" it to somewhere else on the tree. I am aware that this is just a schematic, of course, but perhaps it's easy to come up with a more accurate version of this figure.

4.) On page 11, line 185, the authors quote Figure 2B, but I think it should rather be Supplementary Figure 5 that should be referenced there. Please check.

5.) On page 15, line 264, the authors write "An SFS-based method for demography inference, Stairway plot 2, performed well without high-recombining regions and even better without high-recombining regions in our simulation study.", but I think there is a typo and it shouldn't be two times "without", right?

6.) The availability of the whole genome sequencing data of 179 blackcaps should be indicated more explicitly using accession codes, even though I understand it has been published elsewhere. But researchers should not have to consult another study to access the data, unless there are good reasons for that indirection, I think.

Reviewer #3 (Comments for the Authors (Required)):

[Note this manuscript was co-reviewed with a PhD student]

Ishigohoka and Liedvogel address two important questions in this manuscript: (1) do regions of high recombination affect ancestral recombination graph based inferences, and (2) does removing these regions reduce that effect and improve demographic inference? They start with a clear explanation of why we would expect recombination to affect inference under the ARG, before moving on to simulation studies where they set the ends of the simulated chromosomes to have different ratios of recombination to mutation rates. They show that - in the absence of masking high-recombination regions - inferences of both population size and split time deteriorate with increasing recombination rates. When high-recombining regions are masked, results are (as expected) similar to a scenario with low recombination, providing a practical and relatively simple path to solving the problem where a recombination rate map is available. They then perform a case study in the blackcap, where they partition the genome into low and high-recombining halves, showing that population size and split time inference by MSMC2 (but not Relate) is adversely affected in areas of higher recombination.

This is a simple but well-designed study that clearly answers the questions it sets out to. However, the choice of simulation parameters makes it difficult to assess how well results will generalise. Additionally, they have chosen to test their findings in a species that lacks a proper recombination rate map. If these factors were rectified (or at least clearly justified), this would make a strong contribution to Genetics.

Major comments

1. Simulated population size change (and split times) are very recent relative to the starting population size ($0.01N_e$ Generations ago). To what extent does this affect the authors' conclusions, particularly with respect to MSMC2's performance? Many readers will be interested in inferences of change deeper in the past, so it seems an omission not to consider at least one alternative model with more ancient change. In addition, we note that the N_e is very large in general (at least for vertebrates) - the authors should at least discuss whether the absolute N_e could affect their results, as this will affect ρ ($=4N_e r$).

2. As far as we can tell, the recombination rate map for the blackcap was inferred indirectly from LD data using ρ . Given that LD-based estimates of recombination rate can be confounded with local effective population size (e.g. see Gossmann et al 2011 <https://www.ncbi.nlm.nih.gov/pmc/articles/PMC3241429> and refs therein) could this be an issue for the inference? We suggest

that it would strongly improve the study to use an example system in which the crossover recombination rate has been directly inferred using high-resolution linkage mapping. Moreover, the authors should discuss in more detail what thresholds were used to define high and low recombination rate regions, and how these would relate to recombination rates in other species. This is essential if readers hope to replicate the approach in other systems.

Minor comments

3. The Introduction should mention what is known about recombination rate heterogeneity in species other than vertebrates.
4. The authors need to justify smaller sample size used for MSMC - is it equivalent in computational effort to that used for Relate?
5. We suggest mentioning ARGs in the title to clarify that this concern applies to a specific class of inference tools.
6. In both the study design figure and results plots, the x-axis appears to be truncated at 800 generations before present. This needs to be justified in the text.
7. The cross-coalescence results for Relate are odd when recombination is high (Figure 4 B and C). The authors mention this pattern, but more discussion of its potential cause would be useful.
8. Empirical analysis shows that Relate makes very similar inferences for both high-and low-recombination regions. This result feels a little 'hidden' in the supplementary results, and should be discussed in the Discussion.
9. We suggest including a more classic tree-like representation of the simulation model, (see for example the Demesdraw "tubes" representation <https://github.com/grahamgower/demesdraw> .
10. We recommend separating the code from data in the repository, or (preferably) providing the code through Github or similar.

Associate Editor Comments:

Dear Professor Coop and Reviewers,

We appreciate your engagement in reviewing our manuscript and constructive feedback. Please find our replies to the comments below. We hope the revision has dealt with the points raised for improvements. The referred line numbers in our replies correspond to the PDF without track change. We look forward to further correspondence.

Best regards,
Jun Ishigohoka

GENETICS-2024-306830

High-recombining genomic regions affect demography inference

Dear Dr. Ishigohoka:

Three experts in the field have reviewed your manuscript, and I have read it as well. The reviewers appreciated the work highlighting the issue of haplotype-based methods in high recombination regions. While your manuscript is not currently acceptable for publication in GENETICS, we would welcome a substantially revised manuscript. Both reviewers have comments and concerns to be addressed in a revised manuscript. You can read their reviews at the end of this email.

The reviewers have a number of good recommendations about the simulations. While you do not have to run all these additional simulation, however, you should provide a detailed comments in response. In particular you should justify your choice of parameters more, and likely run simulations with more ancient population size change. We look forward to receiving your revised manuscript. Please let the editorial office know approximately how long you expect to need for revisions.

Upon resubmission, please include:

1. A clean version of your manuscript;
2. A marked version of your manuscript in which you highlight significant revisions carried out in response to the major points raised by the editor/reviewers (track changes is acceptable if preferred);
3. A detailed response to the editor's/reviewers' feedback and to the concerns listed above. Please reference line numbers in this response to aid the editor and reviewers.

Your paper will likely be sent back out for review.

Additionally, please ensure that your resubmission is formatted for GENETICS
<https://academic.oup.com/genetics/pages/general-instructions>

Follow this link to submit the revised manuscript: Link Not Available

Sincerely,

Graham Coop
Associate Editor
GENETICS

Approved by:
David Begun
Senior Editor
GENETICS

Reviewer #1 (Comments for the Authors (Required)):

The manuscript submitted by Ishigohoka and Liedvogel with their evaluation of methods that approximate the ARG with the SMC for demography inference is somehow relevant and might appeal to some Genetics readers. I enjoyed reading this manuscript, especially the guidelines on how to perform demography inference using methods based on ARG in organisms with high recombination rates relative to mutation rates.

But, given the scope of their investigation, as it is presented now, it adds marginally to what was done before in terms of theory and method evaluation of ARG-based methods. Lewanski et al. (2024) review on ARG addressed some problems inherent in the ARG inference framework, especially the one pointed out by the authors (when $\rho > \theta$). It is a problem of identifiability: high rates of recombination relative to the rates of mutation, make the underlying ARG unidentifiable (Hayman et al. 2023), consequently, $\rho > \theta$ affect any parameter computation (TMRCA and N_e). Soni et al. (2024) addressed the rate variability of mutation and recombination rates on the inference of the demographic model and parameters and DFE from the site-frequency spectrum (which seems to be more robust to $\rho > \theta$ as accessed by the present manuscript). We already know that linked selection messes up everything, especially in the form of background selection (Johri et al. 2021), causing the identification of the wrong demographic model and the wrong inference of DFE. Rate variability makes things even worse. Solutions to use recombination (and mutation maps) when developing the null model for inference have already been proposed by Johri et al. (2021, 2022), in the context of rate variability and background selection.

While their simulations as it is now add a bit to our understanding of the impact of high recombination rate on demographic inference, I think their recommendations to empiricists are relevant and somehow lacking in the literature. High recombination rates are a problem, but if you have the right resources you can alleviate it. It is important to show with simulations and analysis of data how to do it properly. It might be a good idea to show how to deal with variable recombination rates, for example, when having one good assembly. With this, we can have an estimation of the recombination map (for example, using Barroso et al. 2019), and use it to remove SNPs in high recombining regions before performing the demographic inference. I think this manuscript would attract empiricists if expand on a little bit on the recommendation side, for example, showing how errors in the inference of recombination maps might affect the inference.

Given this things, I am not sure if Genetics would be an appropriate place for this manuscript as it is presented now. I think there are two main ways that the authors could follow them to make their work more impactful:

1) make the work more general on how higher rates of recombination relative to the mutation rates impact the inference of demography with SMC-based methods; here we should see efforts to make the simulations easy to tease apart each confounding evolutionary force (see more details on the

major reviews): start with models of simple demography (to assess the impact on the inference of population size changes). Incorporate more complex structures, like gene flow or population size changes, but make sure to isolate the impact of each. I would also suggest the incorporation of background selection, although I am aware of the amount of work if it would be the case: learning new things and implementing them. But there are many resources out there to help with this. It would be cool to see the results for when you have islands of high recombination and background selection (see more on this below). If some of those things were accomplished, it would appeal to a broader audience of Genetics;

2) If the authors preferred to keep the narrative of non-model organisms, I think more work should be done on the recommendation part. Indeed, the species that the authors presented empirical results is not a model organism, but they had information to use to mitigate the problem with high rates of recombination that most empiricists don't have. What would be then the recommendations for them? If those things could be incorporated, I think it would appeal to an audience of a more ecological Journal, like Molecular Ecology Resources, for example.

****Major reviews****

First, I think the work might benefit if it started showing results for simulations of simpler demographic scenarios to assess only how high recombination rates might affect the inference. It can be done for example, by simulating only one population with constant size with genomic islands of $\rho/\theta \gg 1$ spaced by regions with where $\rho/\theta = 1$. You also could simulate a genomic with a constant r where $\rho/\theta \gg 1$ (or you can sample many random values of r from a uniform distribution, making it vary quite a lot throughout the genome). Having set the baseline for comparison, you can start making the demographic models more complex, for example with 2-epochs with sudden population size increase/decrease. Finally, compare models where you include exponential size changes with varying (high) recombination rates. For the more complex demographic models, I suggest simulating a control where the $\rho/\theta = 1$.

Reply:

Thank you for the careful assessment of our manuscript and the constructive suggestions. We respond to the comments on the demography scenario and recombination maps separately below.

(1) The suggestion regarding the demography scenario was to start from simpler ones with one population. We decided against adding further simulations in our revision, because although our demography scenario consists of three subpopulations (pop1, pop2, and pop3), we did not include gene flow among the subpopulations after the split event. Therefore, the ancestral process within the sample of each subpopulation is independent. This means, for instance, if we focus on pop1, the simulated data and inference of historical effective population size should be equivalent to your first suggestion (one exception is Relate, where ARG was inferred using 150 diploids from the three populations.). Furthermore, the fact that our result show that MSMC2 and Relate infer the expected demography in the scenario without high-recombining regions (shown in Fig 3A & Fig 4A in our initial submission) also indicates that our chosen demography scenario is simple enough to compare between scenarios with different recombination maps.

(2) We unfortunately could not understand what issues were raised to our recombination maps and how the suggested maps would address them. To clarify what we did; within each of step-wise, narrow, and uniform map types, the five maps share the same shape of recombination landscape, which makes our analysis directly comparable. We set the baseline level of recombination rate as 1/10 the mutation rate, instead of the suggested baseline of the same level to the mutation rate, because we believe this should be a value where ARG-based methods should definitely work. In our

manuscript, the case without high-recombining regions was used as a functional equivalent to the suggested control. Specifically, your suggested control case of uniform recombination map with rate equal to the mutation rate is shown among our uniform maps (see initial submission Supplementary Figure 5B and Supplementary Figure 6 [revision 1 Supplementary Figure 1]).

I think the manuscript needs a better organization, especially the methods and results parts, which seem to overlap and diverge quite a lot. I found many sentences in the results that should be in the methods, for example, if the methods were sufficient, you don't need to write again what you did in the results part.

Finally, I think you should get rid of the "non" model organisms argument from your manuscript. Although it is somehow true that model assumptions for SMC methods are around the idea of $\rho/\theta = 1$, which is, coincidentally, the case for humans, there is nothing in the assumption specific to a model organism. I suggest instead talking about organisms or genomes with elevated recombination rates relative to the mutation rate (it might be that the genome, in general, is like this or some islands are like this).

Reply:

Thank you for the comments and suggestion on how to more comprehensively organise and represent our study. In the revised manuscript, we avoid the dichotomous view of model vs non-model organisms, but instead we mention that ARG-based methods can be affected in some organisms due to their characteristic recombination landscapes (revision 1 L44-66).

It would be nice to see (and maybe, more important to the genetics readers) how genomic regions with $\rho/\theta \gg 1$ interacting with background selection affect the demographic inference with SMC-based methods. I know some of these methods are recommended to use a set of neutral sites. But for organisms where the genome is rich in functional content, this is almost impossible. We know that BGS misleads the inference to population growth, but the investigation where done around with simulation $\rho/\theta = 1$ (Johri et al. 2021). I am wondering what high local recombination rates would do to regions where selection is strong. As pointed out by Cousins et al. (2024), when the strength of BGS increases, the distribution of TMRCA gets younger. Perhaps high recombination rates would decouple mutation from the linked selection? I know it would include more work, but with SLiM it is easy to set up simulations like these.

Reply:

Thank you for the suggestion. Although admittedly beyond the original scope of our study, we were also intrigued by how the effect of the presence of high-recombining regions on demography interact with the effect of background selection. More specifically, we were interested in whether the presence of high-recombining regions, which we illustrate to be problematic for ARG-based demography inference under neutrality, are affected less by background selection than without high-recombining regions. To this end, we performed forward simulations with SLiM, pslim and msprime using the same demography scenario, but with varying ancestral population size of $1e5$ instead of $1e6$ (c.f. response to Reviewer 2 and 3 and revision 1 L123-155 & 397-409 L) with the same recombination maps as the original version, in five scenarios of DFE comparable to Johri et al 2021, ranging from neutral (as a control to confirm consistency with coalescent simulation) to strong background selection. The effect of background selection on demography inference by MSMC2 and Relate was consistent with Johri et al., 2021: the inferred effective population size became smaller as more mutations were deleterious; and the inferred time of population split was shifted to more recent time (revision 1 Supplementary Figures 18, 19, 22, 23). The effect of the presence of high-recombining regions was also visible in a consistent manner to the neutral scenario

in the initial submission. The effects by high-recombining regions and background selection appear to be independent, because masking high-recombining regions improved demography inference, but the effect by background selection was consistent. Therefore, masking high-recombining regions is a practical solution specifically to the presence of high-recombining regions irrespective of the effect by background selection.

****Minor reviews****

- Lines 1-2: remove. I don't see the necessity to direct the abstract to non-model organisms. Model organisms also have regions of high recombination relative to mutation rate (see *Drosophila* for example). You can remove it and make it appeal to a broader audience. You have some interesting findings that apply to demography inference broadly.

Reply:

Thank you for your suggestion. We agree and removed the sentence from the revised Abstract (revision 1 L2).

- Lines 10-11: Replace "non-model organisms" for species.

Reply:

The sentence now reads "Despite the prevalence of high-recombining genomic regions in some organisms, such as birds, its effect on demography inference based on ancestral recombination graph has not been well studied." (revision 1 L6-8).

- Line 21: Remove "reference". A reference genome is not a real entity, and here you are talking about a feature of a species genome.

Reply:

Thank you for the comment. The phrasing was corrected accordingly and now reads "Our results suggest that demography inference using ARG-based methods should be carried out with caution when applied in species whose reference genomes contain long stretches of high-recombining regions." (revision 1 L18-20).

- Lines 23-24: "This part isn't that accurate: "Variation in genomes allows inference of historical demographic parameters". Sometimes it is true in some context (when enough data is available) but most of the time it isn't. Perhaps remove this part and connect the first idea of this paragraph with the next paragraph. "Population history affects the patterns of genetic variation which can be inferred with population genomic methods. Many approaches were developed to take advantage of whole-genome data ..."

- Lines 24-37: Might be removed to sell this manuscript to a broader audience. Keep the first phrase as I pointed out in the comment above and go to the next paragraph.

- Lines 35-37: This is interesting but I don't think it should come first in your introduction. Maybe the next paragraph has more interesting information and is directly linked with your question, which as I understand, is to benchmark SMC methods when recombination rate (r) > mutation rate (μ) ($r > \mu$).

Reply:

Thank you for the suggestions. We reorganised the first paragraphs following your recommendation. (revision 1 L22-66).

- Line 41: change chromosome to region (because are recombining regions or segments within chromosomes).

Reply:

Thank you for the comment. We removed the phrase in question. After the restructure of the paragraphs, the corresponding sentence now reads "Some methods for demography inference are based on the ancestral recombination graph (ARG) (Li & Durbin, 2011; Schiffels & Durbin, 2014; Speidel et al., 2019; Terhorst et al., 2017), a structure that describes the full ancestries of sampled genomes (Griffiths & Marjoram, 1997)." The information of recombination is included in the next sentence in revision L26-29 "It essentially consists of a series of marginal genealogical trees changing in the shape and branch lengths along the chromosome, and their breakpoints effectively represent historical recombinations contributing to the sampled genomes "

****Citations****

Barroso, G. V., Puzović, N. & Dutheil, J. Y. Inference of recombination maps from a single pair of genomes and its application to ancient samples. *_Plos Genet_* ****15****, e1008449 (2019).

Cousins, T., Tabin, D., Patterson, N., Reich, D. & Durvasula, A. Accurate inference of population history in the presence of background selection. *_bioRxiv_* 2024.01.18.576291 (2024)
doi:10.1101/2024.01.18.576291.

Hayman, E., Ignatieva, A., & Hein, J. (2023). Recoverability of ancestral recombination graph topologies. *Theoretical Population Biology*, 154, 27-39. <https://doi.org/10.1016/j.tpb.2023.07.004>

Johri, P. *_et al._* The impact of purifying and background selection on the inference of population history: problems and prospects. *_Mol Biol Evol_* (2021) doi:10.1093/molbev/msab050.

Johri, P. *_et al._* Recommendations for improving statistical inference in population genomics. *_Plos Biol_* ****20****, e3001669 (2022).

Lewanski, A. L., Grundle, M. C. & Bradburd, G. S. The era of the ARG: An introduction to ancestral recombination graphs and their significance in empirical evolutionary genomics. *_PLOS Genet._* ****20****, e1011110 (2024).

Reviewer #2 (Comments for the Authors (Required)):

The paper "High-recombining genomic regions affect demography inference" by Ishigohoka and Liedvogel presents a study of how specific methods for demographic inference perform under high recombining regions.

Many modern methods to infer the history of population sizes from genetic data use haplotype information in some way or another. This includes MSMC2 and Relate, the two studies compared here. The problem this paper addresses is that high recombining regions somewhat break the assumptions of these methods, in that they try to infer local haplotype structure to help reconstructing (parts of) the ARG, which may become impossible under high recombination.

The authors show that indeed high-recombining regions pose a problem for these methods, while another method that relies purely on the site frequency spectrum, ignoring phase and haplotypes, is - unsurprisingly - not affected by recombination rates.

I think the paper is very clear and well written overall.

I only have two main comments:

1.) I think the comparison between MSMC2 and Relate is made difficult by the fact that MSMC2 uses only 8 chromosomes, while Relate uses 300 (page 7). The reason for this is of course simply the different limits of the methods, but I wonder whether one should add a fourth column to Figure 2 and a third column to Figure 3, with Relate run on only 8 chromosomes? That way, it would be become clear whether the noisiness is due to insufficient data (as the authors suggest on page 7) or due to some inherent approximations made by the two methods. At the same time it would leave Relate's superior performance on 300 haplotypes apparent to the reader. Independent on whether the authors agree to add this analysis, I would suggest to add the number of analyse haplotypes somehow to the title line above the plots in Figures 3 and 4, so something like (MSMC2 (8 hapl.), Relate (300 hapl.)), to make clear to the reader that what they are looking at is hard to compare.

Reply:

Thank you for your suggestion. In the revision, we performed Relate using 4 diploid samples per population. Because the demography inference step of Relate is based on inferred ARG, and in order to disentangle the two steps, we ran the demography inference step of Relate using 4 diploids per population with two different ARGs: (1) subset of ARG of 150 diploids from three populations, which we have shown to be good enough to infer demography, and (2) ARG inferred from 12 diploids (4 diploids per population). In both cases, the inference of effective population size in the recent past was noisy (revision 1 Supplementary Figure 8), while the effect of high-recombining regions and masking of high-recombining regions was consistent with the results using 150 diploids instead (revision 1 Supplementary Figures 8). This result indicates that the apparent superiority of Relate is indeed due to scalability. The information on sample size for each method was added to the Figure legend (revision 1 Figures 3 and 4. "We used different numbers of diploid samples for the three methods: 4 diploids/population in `MSMC2`, and 50 diploids/population in `Relate` and `Stairwayplot 2`.").

2.) Related to the same point, and perhaps an alternative way to improve comparability: I wonder why the authors chose a demographic history with such a recent focus. Given that MSMC/PSMC mostly cover times much older than that (with the parameters considered here more like 10^5 rather than 10^4 generations ago) it would be quite interesting to better understand the performance under high recombination for population size changes in the deeper past. An example could be one with periodic growths and declines (https://popsim-consortium.github.io/stdpopsim-docs/stable/catalog.html#sec_catalog_homsap_models_zigzag_1s14). This would then also make the comparison of the two methods - even with drastically different amounts of data input - more useful, as they both then operate in their intended parameter range.

Reply:

Thank you for the comment. The issue is indeed that the split time is very recent compared to the population size ($1e4$ generations compared to the ancestral N_e of $1e6$). In the initial submission, we were particularly interested in scenarios with recent splits because of the direct relevance for our study system, the blackcap, for which we knew that the long-term effective population size is between $1e5$ and $1e6$ and the split time is the order of $1e4$. It is, however, reasonable and indeed generally very valuable to include scenarios with older splits, as both you and Reviewer 3 suggested. Because what matters is the time relative to the effective population size, we have added two scenarios keeping the split time of $1e4$ [generations ago] but reducing the ancestral effective population size to $1/10$ and $1/100$ (i.e. ancestral N_e of $1e5$ and $1e4$ diploids). By keeping the split

time constant, scenarios with a smaller ancestral effective population size mean older split times compared to the population size. By performing MSMC2 and Relate, we found consistent effect of high-recombining regions on demography inference with our original result (revision 1 L335-337, 364-366, Supplementary Figures 3, 4, 6, 7).

... and a few more minor comments:

3.) In Figure 1, the schematic, I noticed that the example trees change a bit more drastically from one recombination event to the next than expected. In particular in B, it appears that trees left and right of a recombination break point differ in multiple internal branch lengths. I think this shouldn't be possible: A single recombination event can break an internal branch and "re-glue" it to somewhere else on the tree. I am aware that this is just a schematic, of course, but perhaps it's easy to come up with a more more accurate version of this figure.

Reply:

Thank you for your good eyes and the comment. The initial figure was technically not wrong because it was based on a toy simulation, but the problem was the order of leaves (which were unlabeled in the figure) was indeed not consistent across trees. The figure has been manually corrected accordingly.

4.) On page 11, line 185, the authors quote Figure 2B, but I think it should rather be Supplementary Figure 5 that should be referenced there. Please check.

Reply:

Thank you for the comment. We corrected it (revision 1 L375).

5.) On page 15, line 264, the authors write "An SFS-based method for demography inference, Stairway plot 2, performed well without high-recombining regions and even better without high-recombining regions in our simulation study.", but I think there is a typo and it shouldn't be two times "without", right?

Reply:

Thank you for the comment. You are absolutely right. We corrected it. Now it reads "An SFS-based method for demography inference, `Stairway plot 2`, performed well without high-recombining regions and even better with high-recombining regions in our simulation study." (revision 1 L469-471).

6.) The availability of the whole genome sequencing data of 179 blackcaps should be indicated more explicitly using accession codes, even though I understand it has been published elsewhere. But researchers should not have to consult another study to access the data, unless there are good reasons for that indirection, I think.

Reply:

Thank you for the suggestion. We fully agree and the revised version now includes the NCBI BioProject identifier the data repository where the VCF file is stored (revision 1 L253).

Reviewer #3 (Comments for the Authors (Required)):

[Note this manuscript was co-reviewed with a PhD student]

Ishigohoka and Liedvogel address two important questions in this manuscript: (1) do regions of high recombination affect ancestral recombination graph based inferences, and (2) does removing

these regions reduce that effect and improve demographic inference? They start with a clear explanation of why we would expect recombination to affect inference under the ARG, before moving on to simulation studies where they set the ends of the simulated chromosomes to have different ratios of recombination to mutation rates. They show that - in the absence of masking high-recombination regions - inferences of both population size and split time deteriorate with increasing recombination rates. When high-recombining regions are masked, results are (as expected) similar to a scenario with low recombination, providing a practical and relatively simple path to solving the problem where a recombination rate map is available. They then perform a case study in the blackcap, where they partition the genome into low and high-recombining halves, showing that population size and split time inference by MSMC2 (but not Relate) is adversely affected in areas of higher recombination.

This is a simple but well-designed study that clearly answers the questions it sets out to. However, the choice of simulation parameters makes it difficult to assess how well results will generalise. Additionally, they have chosen to test their findings in a species that lacks a proper recombination rate map. If these factors were rectified (or at least clearly justified), this would make a strong contribution to Genetics.

Major comments

1. Simulated population size change (and split times) are very recent relative to the starting population size (0.01Ne Generations ago). To what extent does this affect the authors' conclusions, particularly with respect to MSMC2's performance? Many readers will be interested in inferences of change deeper in the past, so it seems an omission not to consider at least one alternative model with more ancient change. In addition, we note that the Ne is very large in general (at least for vertebrates) - the authors should at least discuss whether the absolute Ne could affect their results, as this will affect ρ ($=4N\tau$).

Reply:

Thank you for the comment that has also been touched on by Reviewer 2. In the initial submission, we were particularly interested in scenarios with recent splits because of their direct relevance to our focal study system, the blackcap, for which we knew that the long-term effective population size is between $1e5$ and $1e6$ and the split time is the order of $1e4$. It is, however, reasonable and valuable to include scenarios with older splits, as you and Reviewer 2 suggested (also see our reply to the 2.) main point of Reviewer 2). Because what matters is the time relative to the effective population size instead of absolute value of effective population size and time, we added two scenarios keeping the split time of $1e4$ [generations ago] but reducing the ancestral effective population size to $1/10$ and $1/100$ (i.e. ancestral Ne of $1e5$ and $1e4$ diploids). By keeping the split time constant, scenarios with a smaller ancestral effective population size are equivalent to those with older split times relative to the population size. By performing MSMC2 and Relate, we found the effect of high-recombining regions on demography inference consistent with the original result.

2. As far as we can tell, the recombination rate map for the blackcap was inferred indirectly from LD data using pyrho. Given that LD-based estimates of recombination rate can be confounded with local effective population size (e.g. see Gossman et al 2011 <https://www.ncbi.nlm.nih.gov/pmc/articles/PMC3241429> and refs therein) could this be an issue for the inference? We suggest that it would strongly improve the study to use an example system in which the crossover recombination rate has been directly inferred using high-resolution linkage mapping. Moreover, the authors should discuss in more detail what thresholds were used to define

high and low recombination rate regions, and how these would relate to recombination rates in other species. This is essential if readers hope to replicate the approach in other systems.

Reply:

Thank you for the comments. We first emphasize that the purpose of our empirical analysis is to validate that the impact of recombination rate on demography inference demonstrated with simulation *could* occur empirically, rather than providing the best inference focal for the blackcap demography used here, or presenting a recommended pipeline. Our setup represents the most extreme comparison by using two halves of the genome with high and low recombination rates and therefore does not reflect recommendation for readers, but was primarily used to mirror that the simulated scenarios are indeed of empirical relevance.

Having clarified this objective, the point of the comment is that this comparison of demography inference using two halves of the genome may be irrelevant if these two halves actually do not differ in recombination rate. In general, LD-based maps are shown to be positively correlated with linkage maps (Kawakami et al., 2017; Martin et al., 2019), and sperm typing (Tiemann-Boege et al., n.d.), and have been used to study recombination landscapes (Peñalba & Wolf, 2020). Therefore, we believe that the two halves of the genome in our analysis truly represent high- and low-recombining halves.

Although our empirical analysis as presented in the original submission did not offer recommendation for readers per se, following your comment, we now added investigations regarding how to define genomic regions to mask due to high recombination rate in our revised version (revision 1 L384-396, Supplementary Figures 15-17).

Minor comments

3. The Introduction should mention what is known about recombination rate heterogeneity in species other than vertebrates.

Reply:

Thank you for the important suggestion. We restructured the introduction, and now include a sentence regarding the recombination landscape outside vertebrates, referring to papers on recombination maps in insects, fungi, and plants: “Differences in recombination landscape are also common among a wide range of organisms outside vertebrates (Choi et al. 2013; Lam and Keeney 2015; Smukowski Heil et al. 2015; Stukenbrock and Dutheil 2018; Torres et al. 2023; Torres et al. 2023)” (revision 1 L61-63).

4. The authors need to justify smaller sample size used for MSMC - is it equivalent in computational effort to that used for Relate?

Reply:

Thank you for the question, and we agree this needs clarifying. The different sample size between MSMC2 and Relate was due to different scalability of these methods. In the revised version, we performed Relate with smaller sample size (4 diploids per population) for a fairer and more direct comparison. The results (revision 1 L-328-330, revision 1 Supplementary Figure 8) show that the apparent better performance of Relate was indeed due to having larger sample size than MSMC2. Both methods differ in scalability because MSMC2 runs SMC HMM along a chromosome for each of all pairs of input haplotypes, while Relate runs “chromosome painting” HMM for all haplotypes

together to delineate the chromosome into windows within which all haplotypes are compatible to a bifurcating genealogy.

5. We suggest mentioning ARGs in the title to clarify that this concern applies to a specific class of inference tools.

Reply:

Thank you for the suggestion. We have thought of possible variants, and could have “High-recombining genomic regions affect demography inference based on ancestral recombination graphs”. However, we feel it is a little long and “recombination” appears twice, but if this does not bother readers, we would be happy with this title.

6. In both the study design figure and results plots, the x-axis appears to be truncated at 800 generations before present. This needs to be justified in the text.

Reply:

Thank you for the comment. For the study design figure of revision 1, we plotted the demography from more recent time (10 generations ago). For the plots of inferences, we added the following in the methods: “In Figs. 3, 4, inference of demography older than 1,000 generations ago was shown.” (revision 1 L177-179, L190-191, L197-190).

7. The cross-coalescence results for Relate are odd when recombination is high (Figure 4 B and C). The authors mention this pattern, but more discussion of its potential cause would be useful.

Reply:

Thank you for the comment. The “odd” pattern of rCCR by Relate with high rCCR after population split events (forward in time) indicates that lineages are inferred to coalesce uniformly at random irrespective of populations, instead of more structured coalescent by isolated populations. One possible cause is the effect of incomplete lineage sorting (ILS): in high-recombining regions, a haplotype unique to a population can recombine with a haplotype shared between populations, which may confuse the modified Li-and-Stephens’ HMM and/or the tree builder in Relate. This can be particularly problematic in the most recent past, where the expected branch length is short (due to many remaining lineages to coalesce), which make mutations representing such “mosaic” pattern scarce. In fact, in the added simulation study with smaller ancestral N_e (i.e. older split), this effect is milder, though not negligible, which is in line with the fact that ILS is less prevalent for older population splits. Another possibility, which is not mutually exclusive from the previous one, is that the high rCCR may reflect the uniform prior of coalescence rate by the tree building algorithm of Relate due to the lack of information within high-recombining regions. In line with this possibility, the “odd” pattern of inferred rCCR is even stronger when only 4 diploids per population (12 diploids in total) were used for demography inference (revision 1 Supplementary Figures 8).

We keep this attempt to interpret how the observed impact occurs out of the manuscript for the following two reasons. First, it should be fairer to have such interpretation not only for Relate, but also for MSMC2, but we are not able to discuss this with enough confidence and it will become too speculative. Second, an ultimate explanation to the observed impact on Relate and MSMC2 will end up with algorithmic dissection of these methods, which we believe is far beyond our aim.

8. Empirical analysis shows that Relate makes very similar inferences for both high-and low-recombination regions. This result feels a little 'hidden' in the supplementary results, and should be discussed in the Discussion.

Reply:

Thank you for the comment. Revision 1 L431-434 now reads “In contrast, `Relate` was more robust to the difference in recombination rate between the two conditions (Sup. Fig. 14). This difference between `MSMC2` and `Relate` is in line with our simulation i.e. better inference with larger sample size, enabled in `Relate` due to its high scalability.”

9. We suggest including a more classic tree-like representation of the simulation model, (see for example the Demesdraw "tubes" representation <https://github.com/grahamgower/demesdraw> .

Reply:

Thank you for the suggestion. We have decided to keep the original skyline plots to represent historical N_e in this revision. The main reason is that we present the inference in skyline plot so that we can directly overlay the inference on top of the truth for visual comparison, and we thought having a consistent representation between Figures 2 and 3 makes it more comprehensive and is important for readability and clarity. If you feel the tubes representation is absolutely necessary, we would be happy to include it in the supplementary material, but we think that having two different representations in the main figures would make it more difficult to read.

10. We recommend separating the code from data in the repository, or (preferably) providing the code through Github or similar.

Reply:

Thank you for the suggestion. We are aware that it is common to separate codes from data, but we decided to have them together. Due to the size limit, we could upload only key output files, but not all intermediate files. We thought it would be confusing to find scripts for intermediate steps in a separate place. This is also based on our previous experience in having difficulty navigating published data without much documentation. Of course, we can extract text files for pipeline documentation and scripts from Zenodo and upload them to GitHub (we will if this is absolutely necessary), yet we feel it would be redundant to have the same scripts and documentation in two places. Meanwhile, we do not want to remove scripts and documentation from the Zenodo repository because it would make the data unnavigable.

Associate Editor Comments:

Literature cited

Johri, P., Riall, K., Becher, H., Excoffier, L., Charlesworth, B., & Jensen, J. D. (2021). The Impact of Purifying and Background Selection on the Inference of Population History: Problems and Prospects. *Molecular Biology and Evolution*, 38(7), 2986–3003.

<https://doi.org/10.1093/molbev/msab050>

Kawakami, T., Mugal, C. F., Suh, A., Nater, A., Burri, R., Smeds, L., & Ellegren, H. (2017). Whole-genome patterns of linkage disequilibrium across flycatcher populations clarify the causes

and consequences of fine-scale recombination rate variation in birds. *Molecular Ecology*, 26(16), 4158–4172. <https://doi.org/10.1111/mec.14197>

Martin, S. H., Davey, J. W., Salazar, C., & Jiggins, C. D. (2019). Recombination rate variation shapes barriers to introgression across butterfly genomes. *PLoS Biology*, 17(2), 1–28. <https://doi.org/10.1371/journal.pbio.2006288>

Peñalba, J. V., & Wolf, J. B. W. (2020). From molecules to populations: Appreciating and estimating recombination rate variation. *Nature Reviews Genetics*, 21(August), 476–492. <https://doi.org/10.1038/s41576-020-0240-1>

Tiemann-Boege, I., Calabrese, P., Cochran, D. M., Sokol, R., & Arnheim, N. (n.d.). *High-Resolution Recombination Patterns in a Region of Human Chromosome 21 Measured by Sperm Typing*. <https://doi.org/10.1371/journal.pgen.0020070>

December 20, 2024

RE: GENETICS-2024-307706

Dr. Jun Ishigohoka
Max-Planck-Institut für Evolutionsbiologie
MPRG Behavioural Genomics
August-Thienemann-Straße 2
Plön 24306
Germany

Dear Dr. Ishigohoka:

Congratulations! We are delighted to inform you that your manuscript entitled "High-recombining genomic regions affect demography inference based on ancestral recombination graphs" is acceptable for publication in GENETICS. Many thanks for submitting your research to the journal and for your work through the review process. I had a few small comments on the abstract and introduction that I include as notes on the pdf.

To Proceed to Production:

1. Format your article according to GENETICS style, as discussed at <https://academic.oup.com/genetics/pages/general-instructions>, and upload your final files at <https://genetics.msubmit.net>.
2. Your manuscript will be published as-is (unedited-as submitted, reviewed, and accepted) at the GENETICS website as an Advanced Access article and deposited into PubMed shortly after receipt of source files and the completed license to publish. Please notify sourcefiles@thegsajournals.org if you do not wish to publish your article via Advanced Access.
3. We invite you to submit an original color figure related to your paper for consideration as cover art. Please email your submission to the editorial office or upload it with your final files. You can submit a small-sized image for evaluation, and if selected, the final image must be a TIFF file 2513px wide by 3263px high (8.375 by 10.875 inches; resolution of 600ppi). Please avoid graphs and small type.

If you have any questions or encounter any problems while uploading your accepted manuscript files, please email the editorial office at sourcefiles@thegsajournals.org.

Sincerely,

Graham Coop
Associate Editor
GENETICS

Approved by:
David Begun
Senior Editor
GENETICS

note: Please add jnls.author.support@oup.com and genetics.oup@kwgglobal.com (or the domains @oup.com and @kwgglobal.com) to your email program's "safe senders" list. You will be contacted by both at various points during the production process.

Review comments (if applicable):